# Soil Moisture in the Biebrza Wetlands Retrieved from Sentinel-1 Imagery

**Katarzyna Dabrowska-Zielinska [1],\*, Jan Musial [1], Alicja Malinska [1], Maria Budzynska [1], Radoslaw Gurdak [2], Wojciech Kiryla [1], Maciej Bartold [2] and Patryk Grzybowski [1]**

[1] Institute of Geodesy and Cartography, Jacka Kaczmarskiego 27, 02-679 Warsaw, Poland; jan.musial@igik.edu.pl (J.M.); alicja.malinska@igik.edu.pl (A.M.); maria.budzynaka@igik.edu.pl (M.B.); wojciech.kiryla@igik.edu.pl (W.K.); patryk.grzybowski@igik.edu.pl (P.G.)

[2] Department of Geoinformatics, Cartography and Remote Sensing, Faculty of Geography and Regional Studies, University of Warsaw, Krakowskie Przedmiescie 30, 00-927 Warsaw, Poland; radoslaw.gurdak@igik.edu.pl (R.G.); maciej.bartold@igik.edu.pl (M.B.)

\* Correspondence: katarzyna.dabrowska-zielinska@igik.edu.pl; Tel.: +48-22-3291974

**Abstract:** The objective of the study was to estimate soil moisture (SM) from Sentinel-1 (S-1) satellite images acquired over wetlands. The study was carried out during the years 2015–2017 in the Biebrza Wetlands, situated in north-eastern Poland. At the Biebrza Wetlands, two Sentinel-1 validation sites were established, covering grassland and marshland biomes, where a network of 18 stations for soil moisture measurement was deployed. The sites were funded by the European Space Agency (ESA), and the collected measurements are available through the International Soil Moisture Network (ISMN). The SAR data of the Sentinel-1 satellite with VH (vertical transmit and horizontal receive) and VV (vertical transmit and vertical receive) polarization were applied to SM retrieval for a broad range of vegetation and soil moisture conditions. The methodology is based on research into the effect of vegetation on backscatter ($\sigma^\circ$) changes under different soil moisture and Normalized Difference Vegetation Index (NDVI) values. The NDVI was derived from the optical imagery of a MODIS (Moderate Resolution Imaging Spectroradiometer) sensor onboard the Terra satellite. It was found that the state of the vegetation expressed by NDVI can be described by the indices such as the difference between $\sigma^\circ$ VH and VV, or the ratio of $\sigma^\circ$ VV/VH, as calculated from the Sentinel-1 images in the logarithmic domain. The most significant correlation coefficient for soil moisture was found for data that was acquired from the ascending tracks of the Sentinel-1 satellite, characterized by the lowest incidence angle, and SM at a depth of 5 cm. The study demonstrated that the use of the inversion approach, which was applied to the newly developed models using Water Cloud Model (WCM) that includes the derived indices based on S-1, allowed the estimation of SM for wetlands with reasonable accuracy (10 vol. %). The developed soil moisture retrieval algorithms based on S-1 data are suited for wetland ecosystems, where soil moisture values are several times higher than in agricultural areas.

**Keywords:** Sentinel-1 backscatter; polarization; Terra MODIS; NDVI; soil moisture

## 1. Introduction

The soil moisture (SM) is an essential variable in environmental studies related to wetlands as it controls the biophysical processes that influence water, energy, and carbon exchanges. Hence, there is the need for SM constant monitoring. The SAR satellite imagery is an important source to fulfill this objective regardless of cloud cover and, especially in the areas, in which deployment of in situ SM measurements is not possible or economically unprofitable The possibility of using high temporal and

spatial resolution of the Sentinel-1 (S-1) imagery motivated authors to develop the methodology for its retrieval based on backscattering coefficient ($\sigma^\circ$), as calculated from the VH and VV polarizations.

The study was conducted in the Biebrza Wetlands, situated in north-eastern Poland, with a total area of 59,233 ha. The wetlands are unique in Europe for their non-drained floodplains, marshes, and fens, surrounded by a post-glacial landscape [1]. The Biebrza Wetlands were designated as a wetland site of global importance, as part of NATURA 2000, and since 1995 it has been under the protection of the RAMSAR Convention. Changes in soil moisture towards depletion cause peat mineralization, and the release of substantial amounts of carbon into the atmosphere [2,3]. Therefore, monitoring of soil moisture is very important for the management of the wetlands, to prevent peat degradation. The retrieval of soil moisture (SM) estimates by the means of satellite data is of great interest for a wide range of hydrological applications. The demand for operational SM monitoring was raised in numerous studies, and this was emphasized by the Global Climate Observing System (GCOS) by endorsing SM as an Essential Climate Variable (ECV).

Wetlands are often areas of limited access, where field sampling is difficult due to the inaccessible terrain and the seasonally dynamic nature of the area, and therefore satellites can provide information on the types of wetland vegetation and the dynamics of the local water cycle, in which soil moisture is a significant factor. Controlling soil moisture content is essential for the protection of peat-forming plant communities and for slowing down the drying processes against mineralization [4].

There are numerous studies that describe different remote sensing techniques for the assessment of soil moisture; however the SAR data give very good possibility for frequent spatial monitoring because of their independence from the weather conditions. Kornleson and Coulibaly [5] conducted a comprehensive literature review to provide soil moisture retrieval methodology from SAR data. The researchers have proved that microwave backscatter ($\sigma^\circ$) is affected by the moisture and roughness of the canopy-soil layer. It is further affected by satellite sensor configurations such as the incident angle and the electromagnetic wave polarization [6,7]. The strong interactions of the backscatter signal with the soil and vegetation may not be expressed by simple linear functions. Atema and Ulaby [8] proposed a water cloud model (WCM) that characterized vegetation as the cloud, and represented the total backscatter from the canopy as the sum of the contribution of the vegetation $\sigma^\circ_{veg}$, and of the underlying soil $\sigma^\circ_{soil}$. The WCM model was adopted by Dabrowska-Zielinska et al. [9] for agricultural fields. The separation of the soil and vegetation components is not straightforward due to the complex interactions between them, which simultaneously affect SAR backscatter. The signal strongly depends on the type of vegetation, the amount of moisture, and the type of ecosystem [9]. Wetlands are characterized by deep peat layers, and it is not possible to compare agriculture ecosystems to wetlands, which are wet and very different. Thus, the models derived for wetlands have to be treated separately from models that are designated for agriculture soils and agriculture vegetation.

The C-band SAR on board the ERS-1/2 (European Remote Sensing) satellite, and also on board the ENVISAT (ENVIronmental SATellite), and following the Sentinel-1 satellite, has been applied for soil moisture retrieval [5,10]. The researchers used different models to distinguish the influence of vegetation and soil moisture on the microwave signal. Most of the methods that are applied for soil moisture retrieval have been developed for bare soils and agricultural areas [5,11–15], and only a few have been found for natural environments such as wetlands. Mattia et al. [16] and Balenzano et al. [17] present the SMOSAR (Soil MOisture retrieval from multi-temporal SAR data) algorithm for soil moisture retrieval using the multi-temporal SAR data from Sentinel-1 data. Paloscia et al. [18] developed a soil moisture content (SMC) algorithm for Sentinel-1 characteristics, based on an artificial neural network (ANN), which was tested and validated in several test areas in Italy, Australia, and Spain. Also, ANN-based algorithms for the SMC retrieval applying C-band SAR data (ENVISAT/ASAR, Cosmo-SkyMed) have been adapted and presented by Santi et al. [19]. The overview of the retrieval algorithms presented in [19] demonstrated that ANN is a very powerful tool for estimating the soil moisture at both local and global scales. The proposed model simulates the backscatter of the vegetated areas as a function of the soil backscatter, and the vegetation water

content as computed from the NDVI. Kasischke et al. [20] conducted an investigation on the response of the ERS C-band SAR backscatter to variations in soil moisture and surface inundation in Alaskan wetlands, and found a positive correlation between the backscatter and soil moisture in sites that were dominated by herbaceous vegetation cover. Multi-temporal C-band SAR data, HH, and VV polarized, available from ERS-2 and ENVISAT satellites were used by Lang et al. [21] for the investigation of inundations and soil moisture determination at wetlands. Gao et al. [22] presented two methods for the retrieval of soil moisture over irrigated crop fields based on Sentinel-1 data recorded in the VV polarization combined with Sentinel-2 optical data. The first method used minimum and maximum values of backscattering coefficient calculated from Sentinel-1 data, whereas the second one was based on the analysis of backscattering differences on two consecutive acquisition days. With both methods, the Sentinel-1 data was combined with NDVI index computed from Sentinel-2 data. They obtained estimated RMS soil moisture errors of approximately 0.087 $m^3m^{-3}$ and 0.059 $m^3m^{-3}$ for the first and second methods, respectively. El Hajj et al. [23] used a neural network technique to develop an operational method for soil moisture estimates in agricultural areas based on the synergistic use of Sentinel-1 and Sentinel-2 data. They found that VV polarization alone as well as both VV and VH provides better accuracy on the soil moisture calculation than VH alone. The method developed by them could be applied for agricultural plots with an NDVI lower than 0.75 and allows for the soil moisture estimates with an accuracy of approximately 5 vol. %. Baghdadi at al. [24] applied the Water Cloud Model for estimating surface soil moisture of crop fields and grasslands from Sentinel-1/2 data. They simulated the soil contribution (moisture content and surface roughness) applying Integral Equation Model and used NDVI values as the vegetation descriptor. They obtained that the soil contribution to the total radar signal is large in VV polarization when soil moisture is between 5 and 35 vol. %, and NDVI between 0 and 0.8. Tomer et al. [25] developed an algorithm to retrieve surface soil moisture based on the Cumulative Density Function Transformation of multi-temporal RADARSAT-2 backscattering coefficient. The algorithm, which was tested in a semi-arid tropical region in South India and validated with the in situ data showed RMSE of soil moisture estimates ranging from 0.02 to 0.06 $m^3m^{-3}$ depending on soil information used and development of vegetation. Dabrowska-Zielinska et al. [26] conducted an investigation on soil moisture monitoring in the Biebrza Wetlands using Sentinel-1 data, and found, that LAI dominates the influence on σ° when soil moisture is low. They developed models for soil moisture assessment under different wetland vegetation habitat types (non-forest communities) applying VH polarization ($R^2$ = 0.70 to 0.76). There are not many studies for wetlands SM retrieval applying S-1 data, as can be seen from the literature review. Most of the publications refer to agriculture crops or bare soils. The difference and the ratio of the VH and VV backscatter as the proxy of vegetation conditions has been recently studied and published by several researchers. Vreugdenhil et al. [27] examined Sentinel-1 VV and VH backscatter and their ratio VH/VV to monitor crop conditions with special reference to vegetation water content (VWC) of agriculture crop. Greifeneder et al. [28] demonstrated that the ratio of VH/VV calculated from AQUARIUS L-band scatterometer allows a good compensation of vegetation dynamics for the retrieval of soil moisture. Hosseini et al. [29] used RADARSAT-2 to estimate Leaf Area Index (LAI) for corn and soybeans fields. They found high correlation coefficients between ground measured and estimated LAI values, when dual like-cross polarizations were used (either HH–HV or VV–HV). Also, it has been found that RADARSAT-2 (HH-HV) can be used for the retrieval of soil moisture and the total biomass, while RADARSAT-2 (VV-HV) can be used for the retrieval of the biomass of the wheat heads [30].

The aim of this research study was to examine the sensitivity of Sentinel-1 backscatter (σ°) to SM variation under vegetation, as characterized by different biomasses, and to develop the new models for SM retrieval under wetland vegetation cover (non-forest communities), by applying the C-band SAR data VH and VV polarized, which are available from the Sentinel-1 (S-1) satellite. The vegetation biomass was represented by NDVI, which was calculated by applying the Terra MODIS data. The authors present the approach, which applies the SAR indices such as the difference of σ° VH-VV and the ratio VV/VH as vegetation descriptors in SM retrieval using modified version of WCM.

Application of these descriptors, as dual polarization, give better results to separate the influence of vegetation from the soil moisture impact on backscatter. The modification consists in linearization of WCM model applying Least Squares Method.

The authors are motivated to undertake this study due to the lack of operational methods for the monitoring of SM based on Sentinel-1 data in the Central European wetlands areas. The presented study is a new approach to the previous one [26] on SM modelling based on S-1 data. Due to the temporal frequency of the two S-1 satellites' (S-1A and S-1B) acquisitions, it is possible to monitor soil moisture changes every six days with high spatial resolution (10 × 10 m). The results will highlight the contribution of S-1 data to soil moisture assessment, improving hydrological studies carried out in wetlands, which have so far very often been based on in–situ observations.

## 2. Materials and Methods

### 2.1. Study Area

The Biebrza Wetlands holds 25,494 ha of peatlands, much biodiversity in the rich plant habitats, as well as highly diversified fauna, especially for birds [1]. This is still one of the wildest areas in Europe, and one of the areas that has been least destroyed, damaged, or changed by human activity. The Biebrza Wetlands belong to the largest of Poland's National Parks—Biebrza National Park (BNP), which was created on September 9, 1993 [31]. It is located in Podlaskie Voivodeship, northeastern Poland, and it is situated along the Biebrza River. The geographical position of the study area is: UL: N54° E22°10′ and LR: N53°10′ E23°30′. The Biebrza Wetland area is flat with an average altitude of about 105 m above sea level (m a.s.l.). To the north, the altitude increases, reaching approximately 120 m a.s.l. The main river is the Biebrza River, which flows out near the eastern border of Poland. The Biebrza River drainage basin area is 7051 km$^2$, the river length is 155 km, and its mean flow is 35.3 m$^3$ s$^{-1}$. The Wetlands are flooded annually in the spring, and besides precipitation, flooding is the main supply of moisture into the peat soil. The weather in the Biebrza River Valley is one of the coolest in Poland—the mean year daily temperature is 6.5 °C. The mean sum of the yearly precipitation ranges between 550–650 mm, and is one of the lowest in Poland. The length of the growing season is less than 200 days, and this is one of the shortest in Poland. Generally, summer is warm but short; winter is cold and long. The coldest month is January, with a mean temperature of −4.2 °C, and with temperatures dropping as low as −50 °C. Snow cover can last up to 140 days. July is the warmest month in the Biebrza Valley, with mean temperatures of 17.5 °C, and with temperatures increasing up to 35.3 °C. The length of the summer ranges between 77–85 days [32].

At the Biebrza Wetlands, two sites for Sentinel-1 (S-1) soil moisture (SM) retrieval were established (grassland and marshland), where a network of soil moisture ground stations was built (Figure 1).

Both sites had a flat topography and homogeneous land cover, which ensured the representativeness of average SM estimates across the sites. The environmental conditions between both sites varied with respect to the SM level, vegetation density, and the type of vegetation community cover. The soil moisture for these two sites differed. For the same years, the SM median for the grassland site was equal to 35 vol. % and it was much higher for the marshlands—close to 60 vol. %. The grassland site (Figure 2) was located on an intensively mowed, drained meadow with semi–organic soil (muck-peat soil). The marshland site (Figure 3) was located within the Biebrza National Park, and covered unmanaged sedges with more moist organic soil (peat soil).

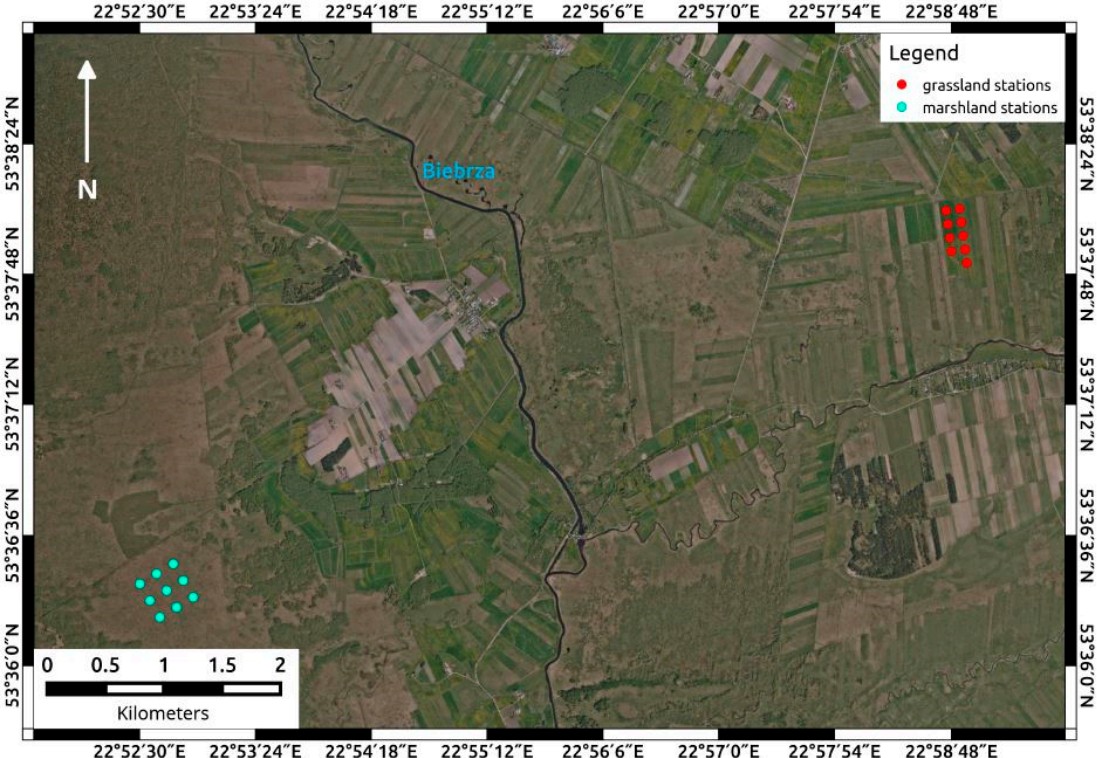

**Figure 1.** Location of S-1 soil moisture sites at the Biebrza Welands overlapped to the Geoportal maps image (www.geoportal.gov.pl).

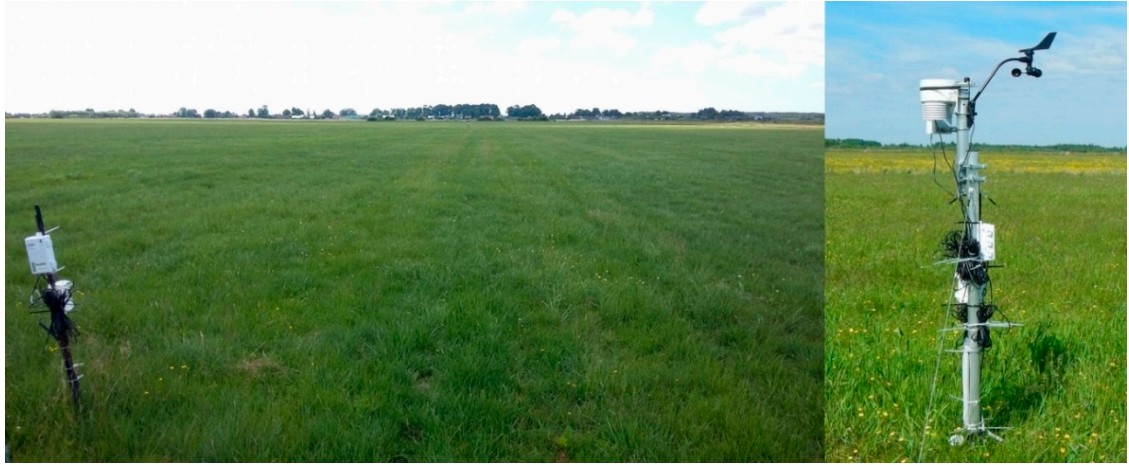

**Figure 2.** Grassland site.

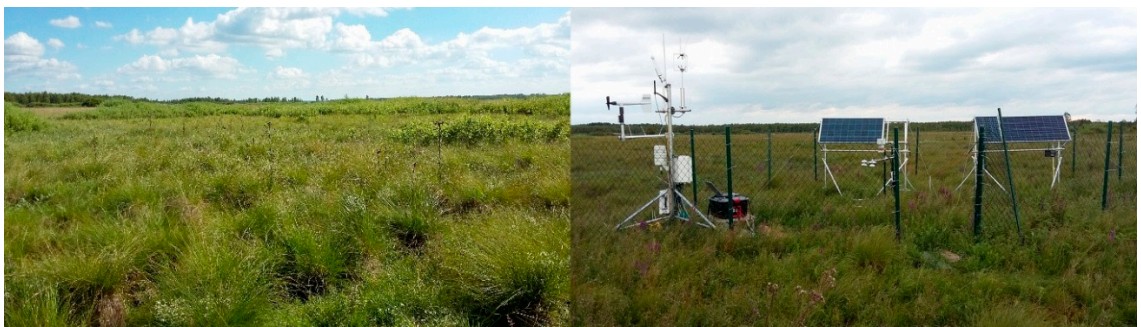

**Figure 3.** Marshland site.

The marshland site had a regular 500 × 500 m measuring grid composed of nine SM stations equipped with five probes each, measuring at the following depths: 5, 10, 20, and 50 cm. The grassland site had analogous instrumentation, with the stations arranged in two rows (230 × 580 m), one with four SM stations, and the second with five SM stations. In total, 90 Decagons GS3 soil moisture sensors were installed.

The grassland and marshland sites featured different soil moisture values and both sites were flooded during the spring. At the marshland site, the water table was very high; therefore, only the soil layer at 5 cm exhibited noticeable variations in water content. The deeper layers were close to saturation point (80–90 vol. %) through the year. An apparent drop of SM values that occurred in winter was related to the ground freezing. At the grassland site, the water table was lower; thus, only the 50 cm soil layer was permanently close to saturation level. The surface soil layers featured a strong annual cycle with a maximum amplitude of around 60 vol. %. A more in-depth description of the sites is available in [33]. The measurements collected from both sites are available through the International Soil Moisture Network (ISMN) [34].

### 2.2. In Situ Data

The in situ data were collected during field campaigns carried out in the years 2015–2017, simultaneous to the satellite overpasses. The positions of the measurement plots were determined using GPS (Global Positioning System). This information was essential for preparing the layer of special measurement points that was needed for the reading and processing of satellite data. Soil moisture (volumetric) was measured by 90 Decagons GS3 sensors calibrated to specific soil conditions at four depths: 5, 10, 20, and 50 cm. The GS3 sensor uses an electromagnetic field to measure the dielectric permittivity of the surrounding medium. The dielectric value is then converted to substrate water content by a calibration equation that is specific to the soil conditions. Regarding the observation modes, the SM measurements were performed every 15 min. Additionally, the height of the vegetation (m) and the biomass wet and dry ($gm^{-2}$) were measured. These data supported the SM analysis with ancillary information about the variables influencing the SAR signal (biomass, vegetation conditions).

During the course of the study, the season of 2015 was extremely dry, whereas conditions in 2017 were extremely wet. In 2016, soil moisture levels were regarded as being moderate.

### 2.3. Satellite Data

Within the study, the following satellite images were used: Sentinel-1 and Terra MODIS. From the SciHUB (Sentinel Scientific Data Hub), Sentinel-1 Level-1 GRDH (Ground Range Detected at High resolution) products, in IWS (Interferometric Wide Swath) acquisition mode (spatial resolution 10 × 10 m) and in a WGS84 ellipsoid, were downloaded. The S-1 images were acquired in the C-band (5.5 GHz) in dual polarization: VV and VH. The nominal acquisition frequency of a single S-1 satellite over the Biebrza Wetlands during the period of the study was 12 days for a single track. However, the grassland site was covered by four different S-1 tracks (two descending and two ascending orbits), and the marshland site was covered by three different S-1 tracks (one descending and two ascending orbits). Furthermore, the availability of the two Sentinel-1A and Sentinel-1B platforms doubled the revisit time, which on average equaled four days for a single satellite and two–three days for two satellites. Table 1 presents the tracks and local incidence angles at the grassland and marshland test sites for selected S-1 relative orbits.

**Table 1.** Local incidence angles for selected S-1 orbit passes (A-ascending, D-descending) and tracks.

| Pass/Track | Marshland Incidence Angle | Grassland Incidence Angle |
|:---:|:---:|:---:|
| A/29 | 43.49° | 43.10° |
| A/131 | 35.59° | 35.13° |
| D/80 | - | 45.65° |
| D/153 | 38.57° | 38.18° |

MODIS images as MOD09Q1 version 6 (V006) products were downloaded from the US Geological Survey website. The MOD09Q1 V006 product provided Bands 1 and 2 (620–670, 841–876, appropriately) at a 250 m resolution in an 8 day gridded level-3 product in the sinusoidal projection. The surface spectral reflectance of Bands 1–2 was corrected for atmospheric conditions such as gasses, aerosols, and Rayleigh scattering. For each pixel, a value was selected from all of the acquisitions within the 8-day composite period, taking into account the cloud coverage and the solar zenith angle [35].

MODIS NDVI 8-day compositions were paired with Sentinel-1 daily satellite images, so that the nearest day of S-1 acquisition to the middle date of 8-day composition of MODIS was taken; therefore, it was assumed that NDVI values could be used to represent the vegetation effect for the modeling of the backscattering coefficients of the S-1. The area of an SM sensors sites is $500 \times 500$ m. The soil moisture, $\sigma°$ and NDVI were taken as the average values for this area.

### 2.4. Methods

Sentinel-1 products were processed with the Sentinel-1 Toolbox (SNAP S1TBX v5.0.4 software) software provided by the European Space Agency (ESA). The processing included: speckle filtering applying a Lee Sigma speckle filter, radiometric calibration, and data conversion to a backscattering coefficient ($\sigma°$) (dB). Then, the scenes were geometrically registered to the local projection PUWG1992, and the $\sigma°$ S-1 values, which corresponded to the measurement sites, were extracted using ERDAS software (Hexagon Geospatial/Intergraph®, Norcross, GA, USA).

The methodology consists of models that were developed for soil moisture retrieval by applying the following Sentinel-1 data: VH and VV polarizations, VH-VV, VV/VH and the NDVI values from the Terra MODIS data. Soil moisture retrieval was based on simplified Water Cloud Model with application of the Least Squares Method.

### 2.4.1. Water Cloud Model with the Least Squares Method

The Water Cloud Model represents the total backscatter from the canopy ($\sigma°$) as the sum of the contribution of the vegetation $\sigma°_{veg}$ and of the underlying soil $\sigma°_{soil}$ [36]:

$$\sigma° = \sigma°_{veg} + \tau^2 \, \sigma°_{soil} \tag{1}$$

where:

$$\sigma°_{veg} = A \, V_1 \cos(\theta) \, (1 - \tau^2) \tag{2}$$

$$\tau^2 = \exp(-2B \, V_2/\cos(\theta)) \tag{3}$$

where: $\theta$—incidence angle, $\tau^2$—two way attenuation through the canopy: $V_1$ and $V_2$ are descriptors of the canopy, A and B are fitted parameters of the model that depend on the vegetation descriptor and the radar configuration. As the vegetation descriptors ($V_1$ and $V_2$), the NDVI values derived from MODIS data were taken. The B parameter is connected with the density of vegetation and its strength of the attenuation during the growing season. For the specific, homogeneous area, we can assume the fixed value of B and apply linearized nonlinear method to solve the WCM model (instead of nonlinear iterative methods). Figure 4 presents the simulation of the strength of attenuation depending on NDVI values for different values of B parameter.

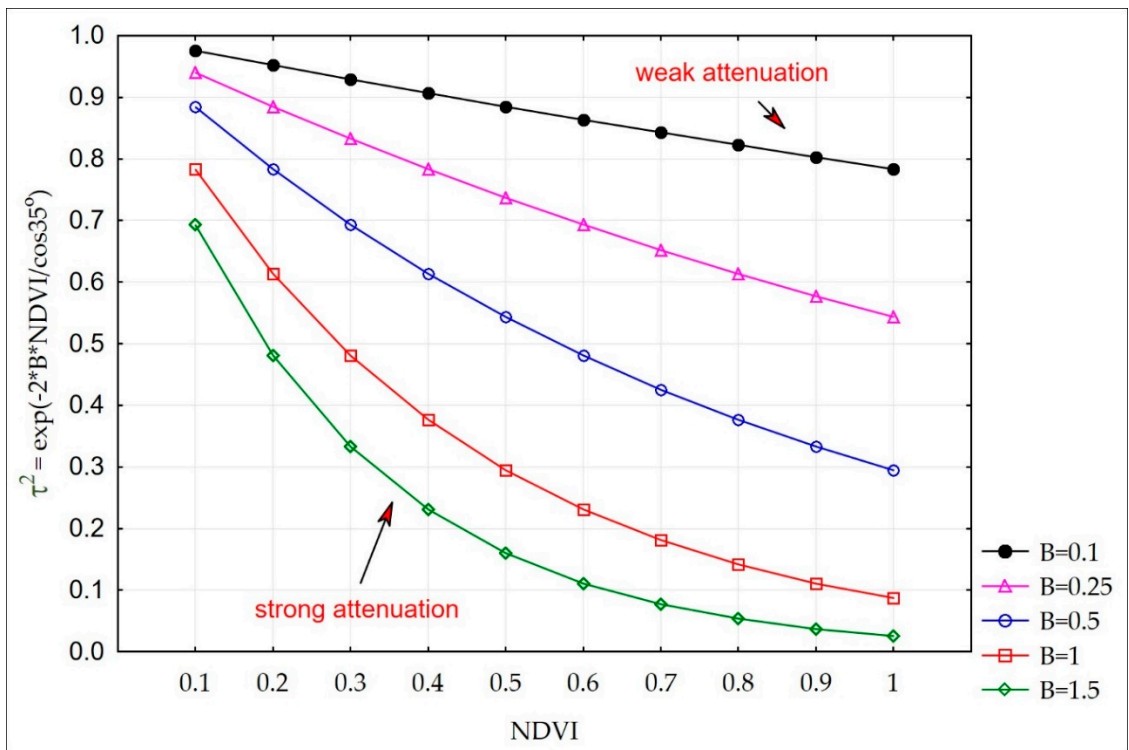

**Figure 4.** Evolution of attenuation ($\tau^2$) depending on NDVI for different values of B parameter.

For bare soil the response of backscatter to soil moisture ($\sigma^\circ{}_{soil}$) is a linear function. It was assumed that in early spring at the wetlands area the soil has dominated impact on backscatter. Therefore we applied modified WCM, where $\sigma^\circ{}_{soil}$ (Equation (1)) was represented by measured SM values. The measurements were conducted during two full years at even time interval, so the relation soil-vegetation can be assumed to be well represented. The following two components of data were designed to describe the effect of the vegetation and the underlying soil on $\sigma^\circ$ VH value: $\tau^2$ * SM, and $(1 - \tau^2)$* cos ($\theta$)* NDVI. The first component represents the interaction of the incident radiation between the vegetation and the underlying soil. $\tau^2$ reduces the impact of the soil on backscatter when the vegetation cover is dense. $\tau^2$ takes the value from 0–1 and is inversely proportional to the vegetation index and to the incidence angle. The second component describes the remaining part of the backscatter that depends on the vegetation canopy covering the soil. The parameters of the model with $\sigma^\circ$ VH as a dependent variable, and $\tau^2$ * SM and $(1 - \tau^2)$* cos ($\theta$)* NDVI as independent variables, were estimated by applying the Least Squares Method. Data were limited to the vegetation season, i.e., from 60–300 days of each year. The form of modified WCM model is the following:

$$\sigma^\circ \text{ VH} = a + b\,\tau^2 \text{ SM} + c(1 - \tau^2)\cos(\theta)\text{ NDVI} \tag{4}$$

where: a, b, c are parameters of regression, that have to be estimated.

2.4.2. Vegetation Descriptors

First, it was assumed that the vegetation index (NDVI) derived from Terra MODIS (described in Section 2.3) could be used as a proxy for the vegetation descriptor of biomass.

Second, the vegetation biomass (expressed by NDVI) was represented by two combinations of sigma VH and sigma VV—the difference and the ratio. This assumption was performed following the approach of using the sigma difference VH−VV as the roughness of the vegetation (in this case, NDVI) following Rao et al. [37]. The $\sigma^\circ$ VH and $\sigma^\circ$ VV values were taken from the processed Sentinel-1 data (described in Section 2.3).

The popular NDVI index works as an indicator that describes the greenness or the density, and the health of the vegetation, based on the measurements of absorption and reflectance. The NDVI was calculated from MODIS MOD09Q1 V006 images on the basis of spectral reflectance from the soil-vegetation surface in the visible red (Band 1) and near-infrared (Band 2) spectra of electromagnetic waves according to:

$$NDVI = (R_{NIR} - R_{RED})/(R_{NIR} + R_{RED}), \tag{5}$$

where: $R_{RED}$—spectral reflectance in the red spectrum, $R_{NIR}$—spectral reflectance in the near-infrared spectrum. For calculating NDVI all pixels with the spectral reflectance values larger than 0 and lower than 10,000 (16 bit unsigned integer) were taken. Then, from Band 3 (Surface Reflectance 250 m State flags) of MOD09Q1 product the pixels flagged as: water, clouds/cloud shadows, and snow/ice were extracted and applied to NDVI images. The values of spectral reflectance were the ratios of the reflected radiation over the incoming radiation in each spectral channel individually (albedo); hence, the NDVI takes on values between 0–1.

### 2.4.3. Statistical Analyses

Statistical analyses were completed in STATISTICA software using the following quality measures: Pearson's correlation, Kendall's tau correlation, R (correlation coefficient), $R^2$ (coefficient of determination), MAPE (Mean Absolute Percentage Error), MPE (Mean Percentage Error), RMSE (Root Mean Square Error), and MBE (Mean Bias Error). The data were checked for the normal distribution and significance prior to all analyses. Validation of the retrieved SM values against the in situ measurements was preformed based on the RMSE error.

## 3. Results

### 3.1. Correlation between σ° Calculated from S-1 and Soil Moisture Measured at Different Depths

The in situ data and satellite data were used in statistical analyses to develop an inversion approach for the estimation of soil moisture from the Sentinel-1 data over the grassland and marshland sites.

Table 2 presents the results of Pearson's correlation (R values) for the marshland site between the backscattering coefficient (σ°) in the polarizations VH and VV, as calculated from Sentinel-1 (S-1), and the soil moisture (SM) when measured in situ at three depths: 5, 10, and 20 cm. The values cover the dates of 26 April 2015 to 30 June 2017. Table 3 presents the same values for grassland site.

**Table 2.** Pearson's correlation (R values) for the marshland site between σ° VH and VV from S-1 and soil moisture (GS3), measured in situ at three depths: 5, 10, and 20 cm.

| Marshland 2015–2017 Pearson Correlation (R) | | | | | | |
| --- | --- | --- | --- | --- | --- | --- |
| Sentinel-1 | | | Soil Moisture GS3 | | | Number of Observations |
| Polarization | Track | Orbit Pass | 5 cm | 10 cm | 20 cm | N |
| VH | 153 | D [1] | 0.49 | 0.34 | 0.40 | 57 |
| | 29 | A [2] | 0.51 | 0.39 | 0.49 | 70 |
| | 131 | A [2] | 0.56 | 0.46 | 0.59 | 66 |
| VV | 153 | D [1] | 0.47 | 0.27 | 0.36 | 57 |
| | 29 | A [2] | 0.40 | 0.22 | 0.28 | 70 |
| | 131 | A [2] | 0.55 | 0.39 | 0.52 | 66 |

[1] Descending, [2] Ascending.

The highest correlation was noted for the S-1 track 131 (ascending pass, low local incidence angles) and the soil moisture as measured at a 5 cm depth. The values of the correlation coefficient in any case were not higher than 0.59 for the marshland site and 0.72 for the grassland site.

For further analysis, the orbit pass ascending (A), and the depth of the soil moisture measurements at a 5 cm depth were taken into account (the highest correlation was found for these dataset).

**Table 3.** Pearson's correlation (R values) for the grassland site between σ° VH and VV from S-1, and soil moisture (GS3) measured in situ at three depths: 5, 10, and 20 cm.

| Grassland 2015–2017 Pearson Correlation (R) | | | | | | |
|---|---|---|---|---|---|---|
| Sentinel-1 | | | Soil Moisture GS3 | | | Number of Observations |
| Polarization | Track | Orbit Pass | 5 cm | 10 cm | 20 cm | N |
| VH | 153 | D [1] | 0.48 | 0.48 | 0.48 | 67 |
| | 29 | A [2] | 0.47 | 0.49 | 0.49 | 79 |
| | 80 | D [1] | 0.28 | 0.29 | 0.27 | 73 |
| | 131 | A [2] | 0.55 | 0.53 | 0.47 | 72 |
| VV | 153 | D [1] | 0.54 | 0.53 | 0.46 | 67 |
| | 29 | A [2] | 0.58 | 0.58 | 0.50 | 79 |
| | 80 | D [1] | 0.39 | 0.37 | 0.26 | 73 |
| | 131 | A [2] | 0.72 | 0.69 | 0.55 | 72 |

[1] Descending, [2] Ascending.

### 3.2. Impact of Vegetation on σ° Calculated from S-1 under Different Soil Moisture Conditions

It was noted that there was a different contribution from the vegetation, as represented by the NDVI, when dry conditions (SM < 30 vol. %) or moist conditions (SM > 60 vol. %) occurred. Figures 5 and 6 show the results of the statistical analyses that were performed between the backscattering coefficient (σ°) value as calculated from VH, and the NDVI as calculated from MODIS for the grassland site. Figure 5 presents the relationship between the σ° value and the NDVI for high, i.e., SM > 60 vol. %, soil moisture when measured at a 5 cm depth. In this case, the vegetation played a role in the process of attenuation when the wave penetrated the vegetation to reach the soil. A different situation was observed when the soil was dry, i.e., SM < 30 vol. %, at a 5 cm depth (Figure 6). The impact of vegetation on the σ° VH was stronger than the impact of soil moisture. Higher biomass values were represented by the NDVI, and hence a higher amount of vegetation moisture content dominated the influence of vegetation on the σ° values. Under low SM conditions, an increase in the NDVI values caused an increase in the σ° VH values, as vegetation impact on backscatter dominates. Under high SM conditions, the vegetation plays the role in two way attenuation of the beam (Equation (3)), an increase of NDVI values caused a decrease in the σ° VH values.

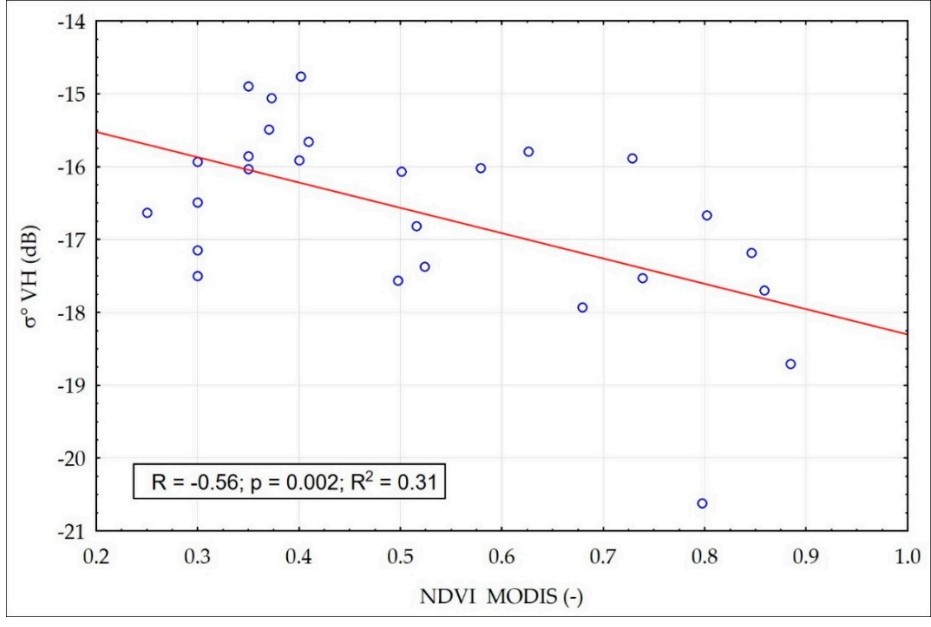

**Figure 5.** Relationship between the NDVI and σ° VH for the SM values measured at a 5 cm depth > 60 vol. % at the grassland site.

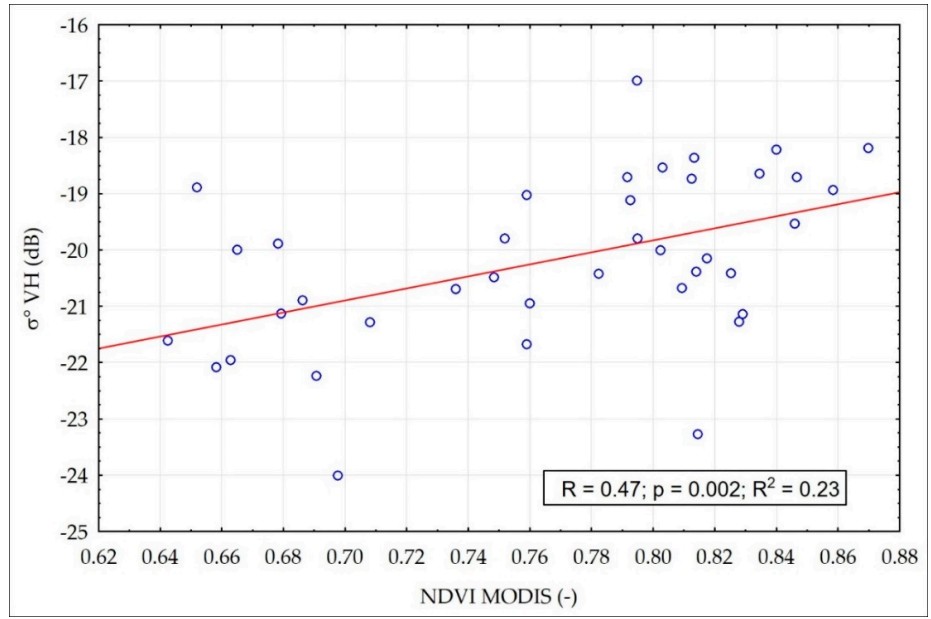

**Figure 6.** Relationship between the NDVI and σ° VH for the SM values measured at a 5 cm depth < 30 vol. % at the grassland site.

### 3.3. Impact of Soil Moisture on σ° Calculated from S-1 under a Quasi-Constant NDVI

If the amount of marshland/grassland vegetation biomass represented by the NDVI is constant in time, the variability of σ° S-1 is consistent with the variability of the soil moisture. Experimental data for the NDVI were gathered for each month separately, and the regression equation between the SM that was measured at a 5 cm depth, and σ° S-1 of the growing season (March–October) was estimated. The obtained correlation coefficients between the soil moisture, and σ° VH and VV were high (Table 4). It was assumed that during the month, the vegetation biomass did not vary significantly, which was confirmed by the low standard deviations values (Table 4) for the NDVI for the particular months. Therefore, it can be assumed that the variability of the backscatter responds to the variability of the soil moisture in areas with homogeneous vegetation cover. However the correlation is significant with the best correlation coefficient (R) for April, May, and October. For the rest of the month the correlation is poor but still significant.

**Table 4.** Correlations between σ° VH and VV and SM at a 5 cm depth for the grassland and marshland sites during the seasons of 2015–2016.

| Month | NDVI | | SM and σ° VH | | | SM and σ° VV | | | N [4] |
|---|---|---|---|---|---|---|---|---|---|
| | Mean | SD [1] | R [2] | *p*-Value | S [3] dB/Vol.% | R [2] | *p*-Value | S [3] dB/Vol.% | |
| March | 0.43 | 0.08 | 0.87 | 0.00 | 0.24 | 0.64 | 0.02 | 0.14 | 13 |
| April | 0.44 | 0.10 | 0.83 | 0.00 | 0.14 | 0.86 | 0.00 | 0.12 | 30 |
| May | 0.63 | 0.16 | 0.87 | 0.00 | 0.10 | 0.85 | 0.00 | 0.11 | 33 |
| June | 0.78 | 0.08 | 0.58 | 0.00 | 0.04 | 0.18 | 0.35 | - | 29 |
| July | 0.77 | 0.06 | 0.61 | 0.00 | 0.04 | 0.19 | 0.27 | - | 31 |
| August | 0.80 | 0.08 | 0.53 | 0.00 | 0.03 | 0.24 | 0.17 | - | 32 |
| September | 0.76 | 0.07 | 0.83 | 0.00 | 0.06 | 0.51 | 0.00 | 0.03 | 35 |
| October | 0.63 | 0.10 | 0.86 | 0.00 | 0.07 | 0.80 | 0.00 | 0.06 | 41 |

[1] Standard deviations, [2] Correlation coefficient, [3] Sensitivity, [4] Number of observations.

Sensitivity of backscatter to the soil moisture is the measure of the change in σ° with the change in soil moisture. It was defined as the slope of the regression line between them at a given vegetation conditions. The higher values of sensitivity occurred in early spring when vegetation cover was lower than in later part of the growing season (Table 4).

*3.4. Compatibility of Seasonal Trends in the Course of the Vegetation Descriptor NDVI, and the σ° Difference VH−VV and Ratio VV/VH*

The time series of σ° indices that were calculated as the difference of polarization VH−VV, or the ratio VV/VH, presented seasonality trends, i.e., variations that were specific to a particular timeframe. There was a systematic increase of σ° VH−VV and VV/VH values during the growing season, and a decrease in autumn, similar to the behavior of NDVI. Figure 7 presents the temporal evolution of the NDVI and σ° VH−VV values during the vegetation season in 2016 at the grassland test site as an example. Mann-Kendall tau statistics were performed for both sites for the seasons of 2016–2017 separately (two complete growing seasons of observations). It revealed that the compatibility of the seasonal trends of σ° VH−VV and VV/VH with the NDVI were statistically significant (Table 5).

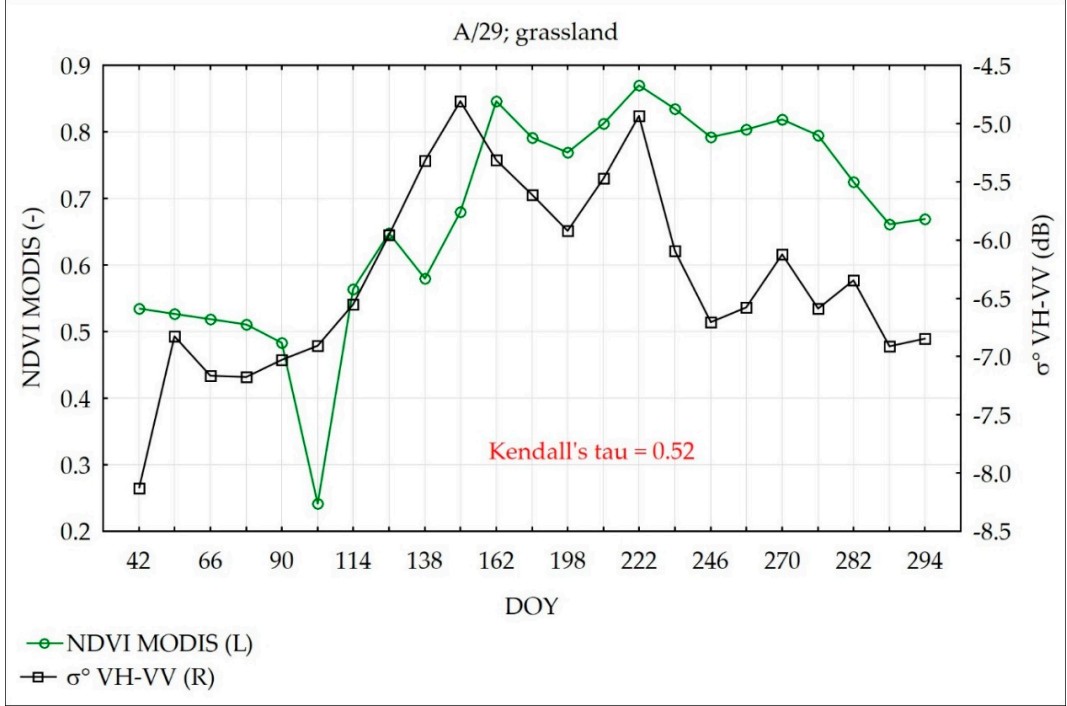

**Figure 7.** Temporal evolution of the NDVI and σ° VH−VV during the vegetation season of 2016 on the grassland site.

**Table 5.** Kendall's tau statistics between the NDVI and the σ° indices VH−VV and VV/VH for the grassland and marshland sites.

| Site | Year | S-1 Track | Kendall's Tau for VH−VV | N [1] | Kendall's Tau for VV/VH |
|---|---|---|---|---|---|
| grassland | 2016 | 29 | 0.52 | 37 | 0.42 |
| | | 131 | 0.39 | 36 | 0.46 |
| | 2017 | 29 | 0.51 | 27 | 0.28 |
| | | 131 | 0.50 | 26 | 0.28 |
| marshland | 2016 | 29 | 0.35 | 37 | 0.37 |
| | | 131 | 0.54 | 36 | 0.56 |
| | 2017 | 29 | 0.68 | 27 | 0.39 |
| | | 131 | 0.74 | 25 | 0.32 |

[1] Number of observations.

Thus, it has been assumed that the influence of vegetation on σ° S-1 values could be expressed by indices of the difference between σ° VH and VV (VH−VV) and the ratio of σ° VV/VH. Analyzing Kendall's tau coefficients for all test sites, tracks, and seasons, it was found that both σ° VH−VV and σ° VV/VH indices were in monotonic correlation with the NDVI, and that they could replace

the NDVI values in soil moisture modeling. In the experiment, the values of σ° VV/VH was always positive and less than 1.

By applying the indices calculated using the S-1 data in modeling SM, the independence from the optical data (often overcast conditions) was ensured. Also, it allowed for quick calculations of soil moisture, which often changes rapidly and has to be observed regularly.

The two following approaches are presented in building the model for soil moisture retrieval:

1   Using the NDVI as a vegetation descriptor
2   Substituting the NDVI by the index σ° VH−VV and the index σ° VV/VH

### 3.5. Soil Moisture Retrieval Using σ° from Sentinel-1 and NDVI from MODIS

Figure 4 shows, that the attenuation of radar signal by vegetation at high moisture conditions of soil was in the range of 3 dB, while the whole range of σ° VH variability was 12 dB. Taking the level of attenuation as a middle, the value of B = 0.5 was chosen for further analysis. Thus, it was assumed that radar signal is attenuated by the vegetation in wetland according to:

$$\tau^2 = \exp(-NDVI/\cos(\theta)) \tag{6}$$

The parameters in (Equation (4)) were estimated as follows:
Model 1a:

$$\sigma° \ VH = -28.3 + 0.2\tau^2 \ SM + 14.7(1 - \tau^2) \cos(\theta) \ NDVI \tag{7}$$

where: R = 0.92; $R^2$ = 0.85; $p < 0.0000$; N = 147; Std. Err. = 0.79 dB, for ascending orbit.

The partial correlations for the soil and vegetation components were 0.89 and 0.54, respectively, which means that soil moisture influenced σ° VH more strongly than the vegetation cover. Figure 8 presents a comparison between the observed values of σ° VH (derived from S-1 images) and those that were predicted using Model 1a (Equation (7)).

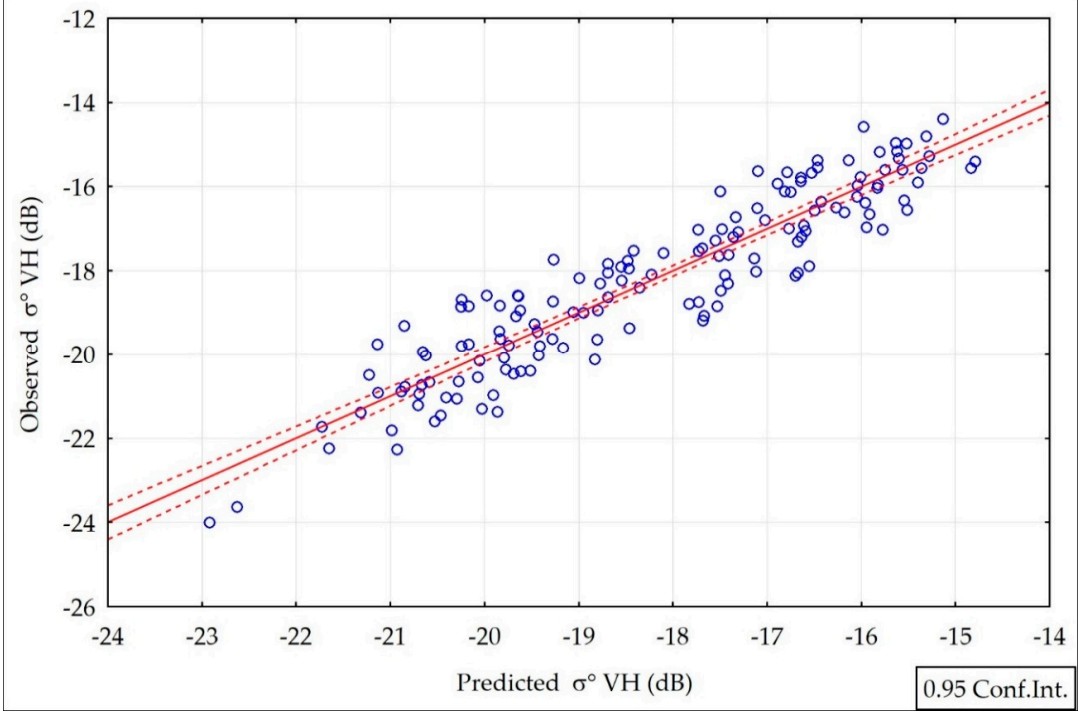

**Figure 8.** The σ° VH values observed and predicted by Model 1a (Equation (7)).

Applying Linear Multiple Regression Model (Equation (4)), three parameters of the WCM model were estimated. Parameter "c" equal to 14.7 in (7) corresponds to "A" in WCM (Equation (2)). The remaining two parameters were interpreted as follows: "b" equal to 0.2 as sensitivity and "a" equal to −28.3 as intercept of SAR backscatter under fixed NDVI = 0 conditions. Intercept is the backscatter value expected for the dry soils. It is mainly a function of surface roughness [38]. For bare soil, where NDVI = 0 (theoretically), what means $\tau^2 = 1$ and $\sigma^\circ_{veg} = 0$, the Equation (7) takes the following form: $\sigma^\circ$ VH = −28.3 + 0.2* SM. For the early spring measurements, when the vegetation has not started yet to grow, estimated equation has the following form: $\sigma^\circ$ VH = −34.4 + 0.21*SM, where R = 0.89; N = 34. In both simulated and estimated equations, the regression slope that means sensitivity, is the same. The intercept parameters which are connected with roughness of soil and vegetation cover, differ. This is the measure of the difference between the soil, theoretically bare, according to model (Equation (7)) and our assumption.

Model 1b:

$$\sigma^\circ \text{ VV} = -21.5 + 0.19\tau^2 \text{ SM} + 12.3(1 - \tau^2) \cos(\theta) \text{ NDVI} \tag{8}$$

where: R = 0.91; $R^2$ = 0.82; $p$ < 0.0000; N = 170; Std. Err. = 0.84 dB, for ascending orbit.

The partial correlation for the soil and vegetation components were 0.87 and 0.50 respectively, which means that soil moisture influenced $\sigma^\circ$ VV more strongly than the vegetation cover. Figure 9 presents a comparison between the $\sigma^\circ$ VV values observed (derived from satellite images) and predicted by Model 1b according to Equation (8).

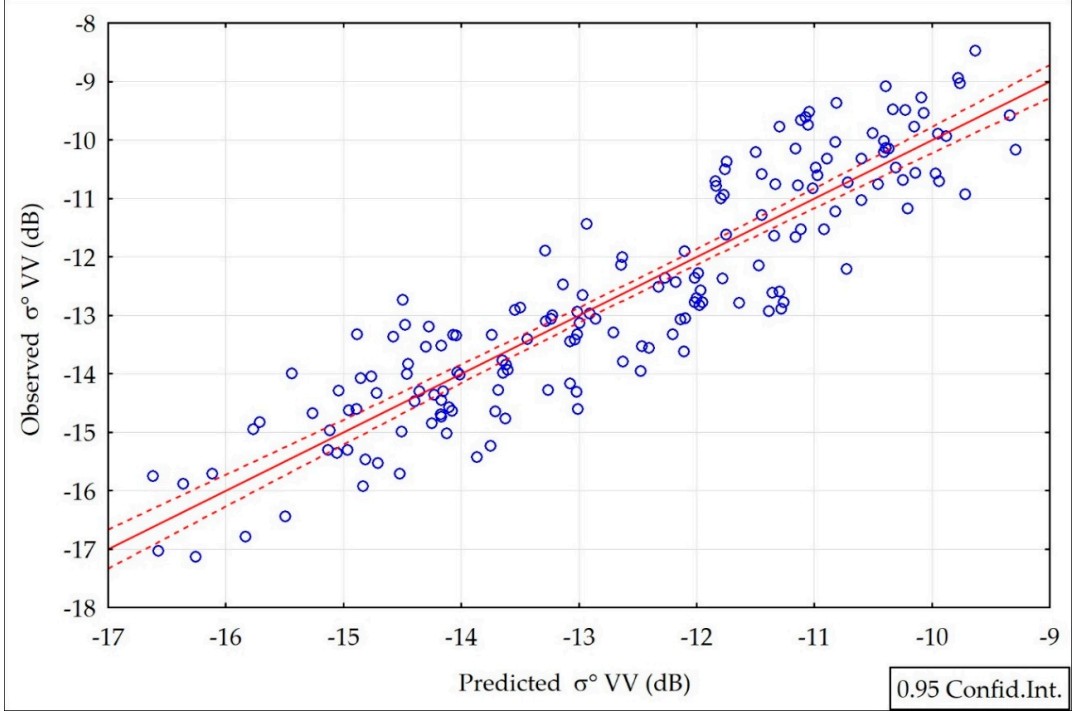

**Figure 9.** The $\sigma^\circ$ VV values observed and predicted by Model 1b (Equation (8)).

The models 1a–1b present the influence of soil moisture and vegetation cover (expressed by NDVI from MODIS) on the S-1 backscatter. The standard errors of estimation for $\sigma^\circ$ VH and $\sigma^\circ$ VV were 0.79 dB and 0.84 dB, respectively.

Table 6 presents the mean absolute percentage errors (MAPE) of the $\sigma^\circ$ S-1 ascending pass, assessed by Model 1a and Model 1b for the years 2015–2017 for the two sites and the two tracks separately. MAPE1 applies to Model 1a, and MAPE2 applies to Model 1b. The mean percentage error for $\sigma^\circ$ VH estimation was 6.6%, and for $\sigma^\circ$ VV estimation, it was 8.8% for all observations (not only the teaching set). The distribution of the error was well balanced on the sites and the tracks.

**Table 6.** Mean absolute percentage error (MAPE) errors of σ° VH and VV derived from Model 1a and Model 1b for the years 2015–2017.

| Site | Track | MAPE1 [1] (%) | MAPE2 [2] (%) | Number of Observations |
|---|---|---|---|---|
| Grassland | 131 | 5.7 | 8.7 | 62 |
| | 29 | 5.9 | 8.8 | 56 |
| Marshland | 131 | 7.6 | 8.8 | 45 |
| | 29 | 7.2 | 8.8 | 47 |
| All | | 6.6 | 8.8 | 200 |

[1] Errors applies to Model 1a, [2] Errors applies to Model 1b.

Figures 10 and 11 present the simulation of σ° VH and σ° VV with the increase of the NDVI for various values of soil moisture from the range of 10–90 vol. %. The increase of σ° with the increase of the NDVI was significant with low soil moisture, the attenuation of the signal was small. When the soil moisture was high, the increase of the NDVI influences the decrease of σ°.

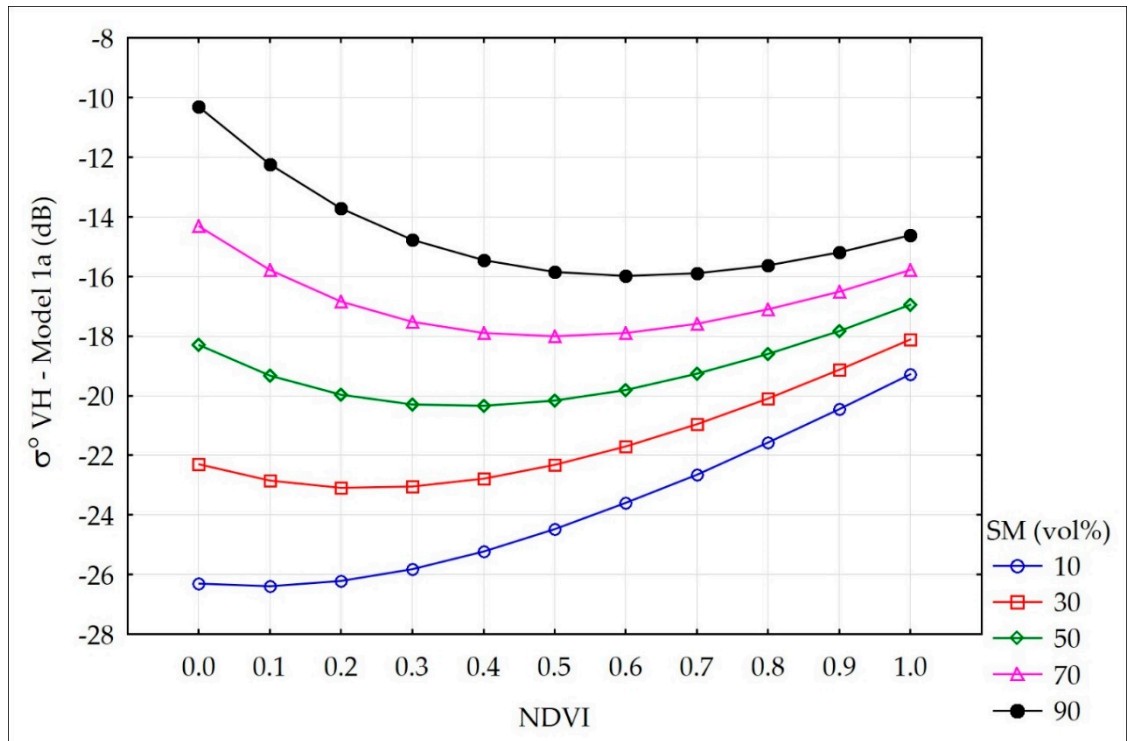

**Figure 10.** Impact of NDVI on σ° VH under various levels of soil moisture (SM) according to Model 1a.

Figures 10 and 11 present the soil and vegetation impact on σ° VV and σ° VH according to Models 1a–1b. The sensitivity of both polarizations on soil moisture under given vegetation condition (NDVI) was similar at wetland area (parameter b in Equations (7) and (8)). Taking the observed range of NDVI as 0.3–0.8, the sensitivity of σ° VH was calculated. For the satellite track 29 ($\theta = 43°10'$) the obtained highest sensitivity was about 0.088 dB/vol. % and the lowest–0.022 dB/vol. %, while for the satellite track 131 ($\theta = 35°13'$) − 0.095 dB/vol. % and 0.028 dB/vol. %, respectively.

The soil moisture can be retrieved through the inversion of Model 1a (Equation (7)) with an accuracy of 9.8 vol. % (Equation (9)). The errors were similar for two sites.

$$\text{SM} = (\sigma° \text{ VH} + 28.3 - 14.7 * (1 - \tau^2) * \cos(\theta) * \text{NDVI})/(0.2 * \tau^2) \tag{9}$$

Table 7 presents the RMSE errors (vol. %) for selected ranges of soil moisture values (5 cm depth) based on Model 1a. It was noted that for the high SM values (in the range of 80–100 vol %) errors were lower than those of the remaining SM ranges.

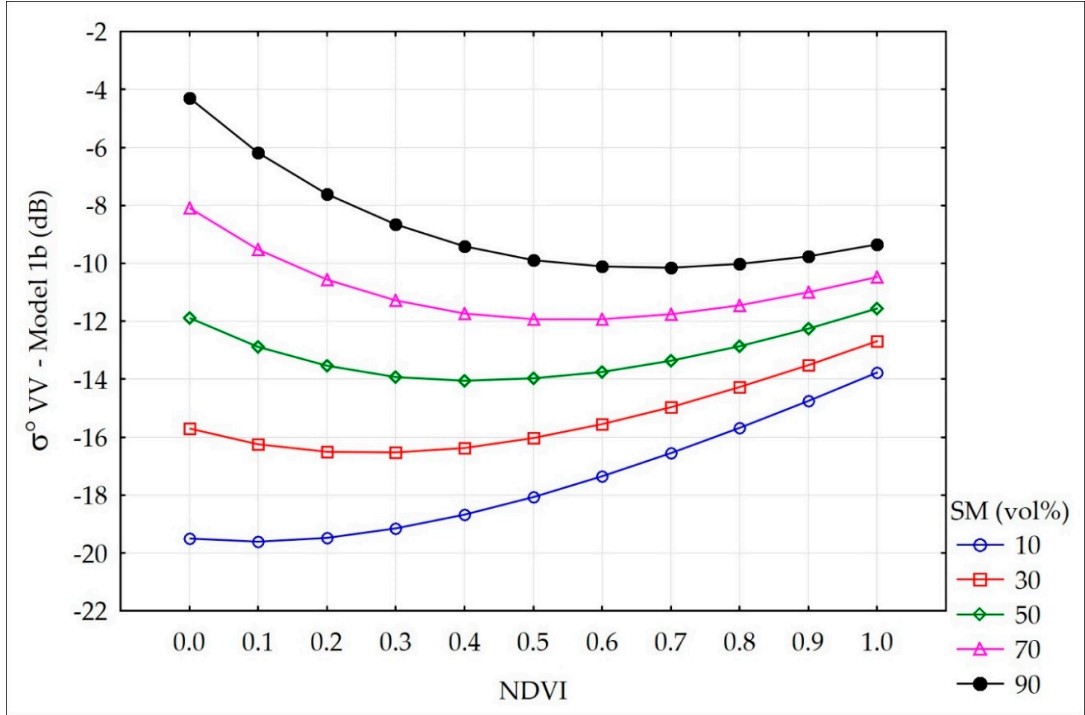

**Figure 11.** Impact of NDVI on σ° VV under various levels of soil moisture (SM) according to Model 1b.

**Table 7.** Errors of soil moisture retrieval by Model 1a (ascending pass) 2015–2017.

| SM Range | N [1] | RMSE (vol. %) |
|---|---|---|
| 20–40 | 39 | 11.8 |
| 40–60 | 39 | 9.5 |
| 60–80 | 35 | 9.9 |
| 80–100 | 34 | 8.4 |
| All | 147 | 9.8 |

[1] Number of observations.

Table 8 presents the RMSE errors (vol. %) for selected ranges of the NDVI values based on Model 1a. The RMSE errors were between 7.4–11.5 vol. %. It was clearly visible that the error was higher with denser vegetation cover (higher NDVI values).

**Table 8.** Errors of soil moisture retrieval by Model 1a for different densities of vegetation.

| NDVI Range | N [1] | RMSE (vol. %) |
|---|---|---|
| 0.2–0.4 | 24 | 7.4 |
| 0.4–0.6 | 40 | 8.5 |
| 0.6–0.8 | 47 | 10.4 |
| 0.8–0.9 | 36 | 11.5 |
| All | 147 | 9.8 |

[1] Number of observations.

### 3.6. Soil Moisture Retrieval Using the σ° Indices from Sentinel-1

Replacing vegetation index NDVI in Equation (4) by σ° VV/VH values we receive:

$$\tau^2 = \exp(-2(\sigma^\circ \text{ VV}/\text{VH})/\cos(\theta)) \tag{10}$$

where: $\sigma^\circ$ VV and VH had the only negative values in our study, and $\sigma^\circ$ VV/VH <1, and B was fixed to 1. The choice of B value was preceded by the same analysis as in the case of Models 1a–1b. Two components were designed to describe the effect of the underlying soil and vegetation on the $\sigma^\circ$ VH value: $\tau^2$*SM and $(1 - \tau^2)$*cos $(\theta)$*$\sigma^\circ$(VH–VV)$^2$. Then, $\sigma^\circ$ VH was modeled according to Model 2 applying linearized nonlinear regression method.

Model 2:

$$\sigma^\circ \text{ VH} = -18.9 + 0.33\tau^2 \text{ SM} - 0.14(1 - \tau^2)\cos(\theta)\,\sigma^\circ(\text{VH-VV})^2 \tag{11}$$

where: R = 0.91; $R^2$ = 0.82; $p <$ 0.000; N = 252; Std. Err. = 0.70 dB, (Figure 12), for ascending orbits.

There is no redundancy of independent components in the multiple regression model. The correlation between them is $R^2$ = 0.002.

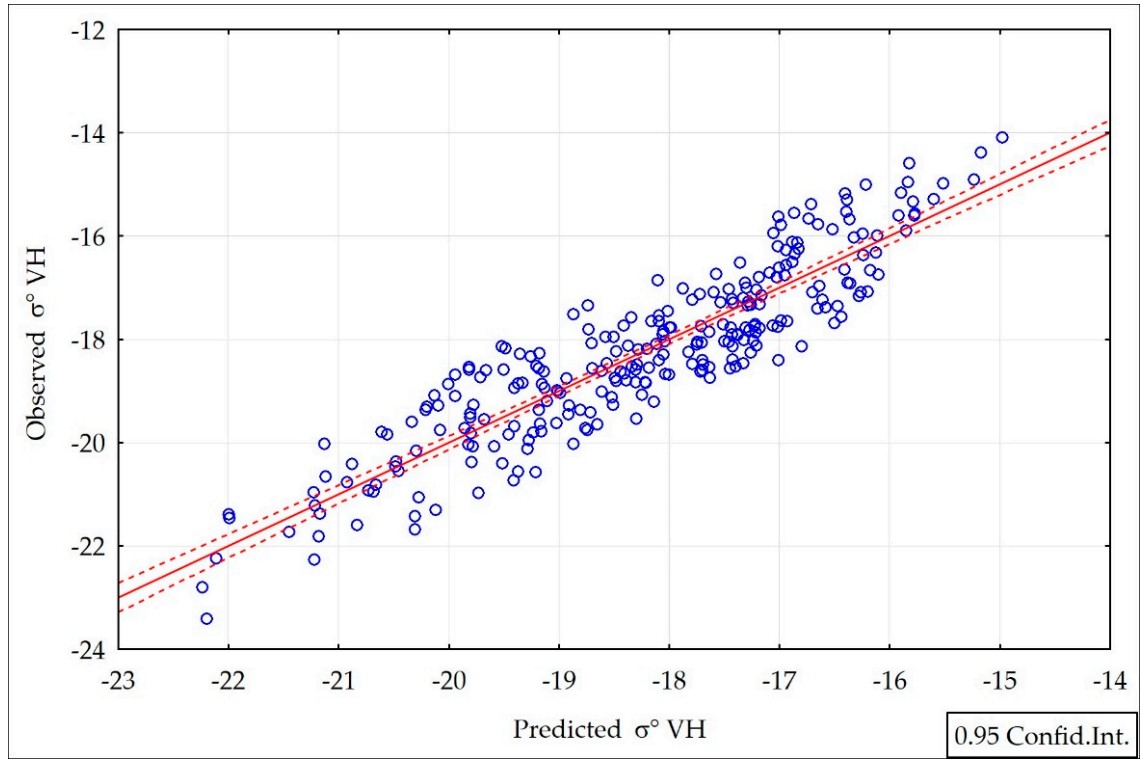

**Figure 12.** The $\sigma^\circ$ VH values observed and predicted by Model 2 (Equation (11)).

Three regression parameters could be interpreted as follows: c = 0.14 as vegetation parameter corresponding to A in Equation (2); b = 0.33 as sensitivity of SAR backscatter for $\tau^2$ = 1; constant a = −18.9 is the state of balance between the impact of vegetation and the underlying soil on $\sigma^\circ$ VH (SM about 50 vol. %, Figure 13). Under $\sigma^\circ$ VV < 0 the attenuation factor $\tau^2$ (Equation (10)) is always less than 1, so the sensitivity does not reaches the value of 0.33, it is lower. Theoretically, sensitivity of SAR backscatter to soil moisture increases when the ratio $\sigma^\circ$ VV/VH decreases. Figure 14 shows the periods under low vegetation conditions.

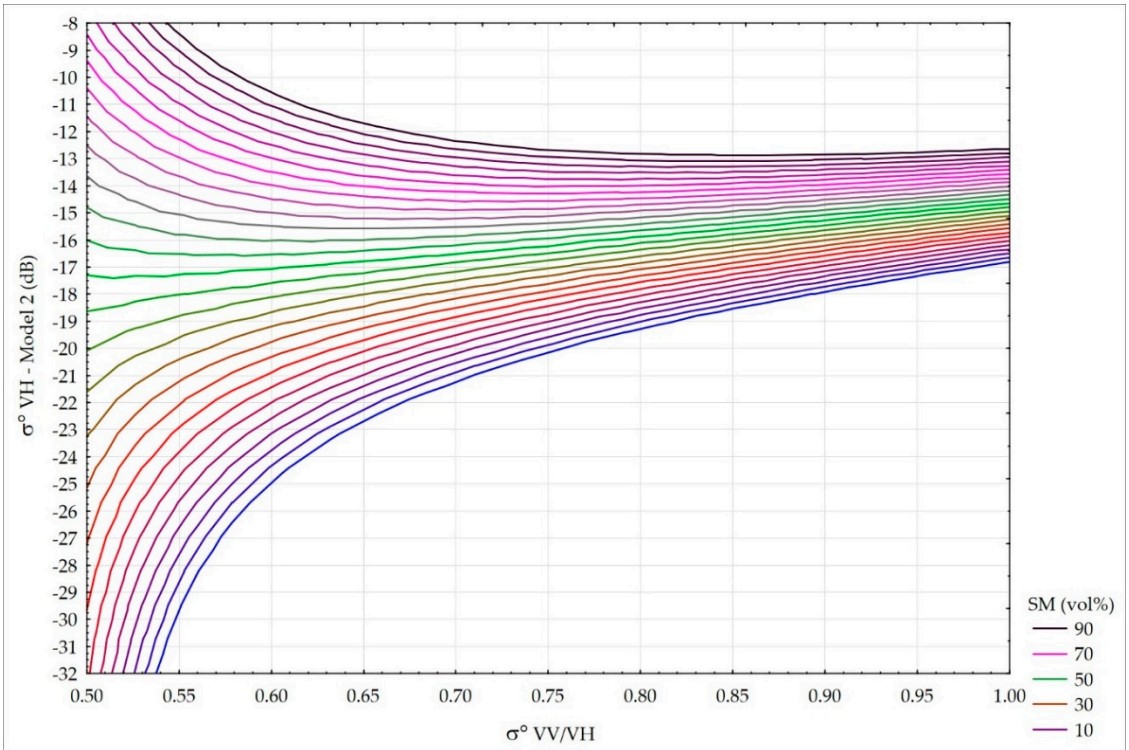

**Figure 13.** Impact of vegetation described by σ° VV/VH on σ° VH for different SM values according to Model 2.

Figure 13 presents the changes of σ° VH sensitivity during the vegetation development represented by σ° VV/VH. Taking the observed range of σ° VV/VH as 0.5–0.9, and $\tau^2$ for each track separately, the range of sensitivity of σ° VH backscatter was calculated. For the satellite track 29 (θ = 43°10′), the highest sensitivity was 0.084 dB/vol. % and the lowest was 0.029 dB/vol. %, while for the satellite track 131 ((θ = 35°13′) − 0.096 dB/vol. % and 0.036 dB/vol. %, respectively. It is compatible with the results when the NDVI from optical data were used (Figures 10 and 11). For low SM there is the increase of σ° VH. For high values of SM, there is the attenuation of the beam by vegetation. Model 2 can be applied in all weather conditions, independently of sky conditions, on which the acquisition of optical images depends.

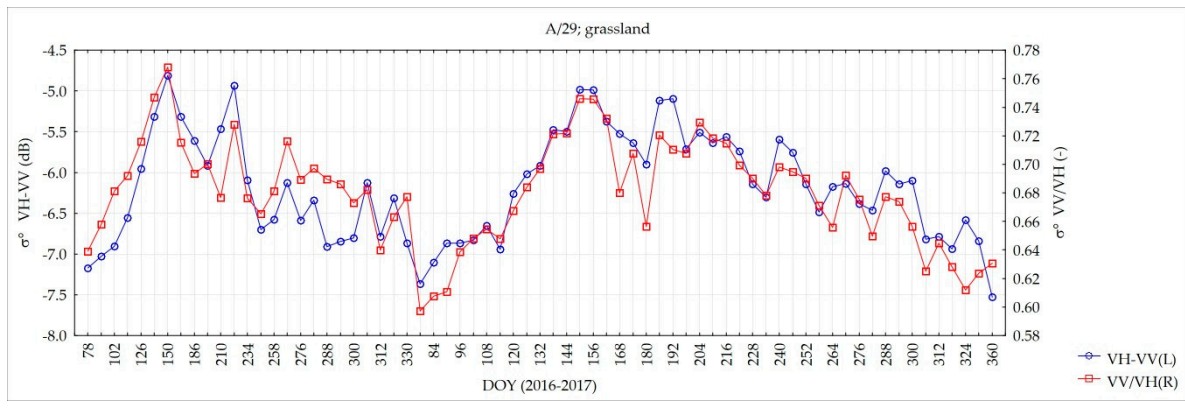

**Figure 14.** Time series of σ° VH-VV and σ° VV/VH during the years 2016–2017 for the grassland site.

From the inversion of Model 2 (Equation (11)), the soil moisture was calculated as follows:

$$\text{SM} = (\sigma° \text{ VH} + 18.9 - 0.14(1 - \tau^2) \cos(\theta) \, \sigma°(\text{VH}-\text{VV})^2 / (0.33 \, \tau^2) \tag{12}$$

The mean RMSE error of the soil moisture retrieved from Model 2 Equation (12) was 13 vol. % (Tables 9 and 10). Table 9 presents the RMSE errors from data for the whole year when the soil temperature is >278 °K. Table 10 presents the RMSE errors for the data from the vegetation season, i.e., from the DOY (Day Of the Year) 60–300.

**Table 9.** Errors analysis for different ranges of SM (5 cm depth) as retrieved by Model 2 (whole year).

| SM Range (vol. %) | N [1] | RMSE (vol. %) |
|---|---|---|
| 20–40 | 51 | 14.8 |
| 40–60 | 53 | 11.8 |
| 60–80 | 64 | 13.8 |
| 80–100 | 40 | 13.6 |
| All | 252 | 13.0 |

[1] Number of observations.

**Table 10.** Errors analysis for different ranges of SM (5 cm depth) as retrieved by Model 2 (growing season).

| SM Range (vol. %) | N [1] | RMSE (vol. %) |
|---|---|---|
| 20–40 | 60 | 12.1 |
| 40–60 | 70 | 12.3 |
| 60–80 | 60 | 14.7 |
| 80–100 | 62 | 11.8 |
| All | 208 | 13.5 |

[1] Number of observations.

The validation of Model 2 was performed for the S-1 data between September 2017–May 2018. The data from December–March were excluded, as the soil temperatures were lower than 278 °K. Table 11 presents the RMSE errors for the data used in the validation procedure.

**Table 11.** Errors analysis for different ranges of SM (5 cm depth) retrieved by Model 2 for validation data.

| SM Range (vol. %) | N [1] | RMSE (vol. %) |
|---|---|---|
| 40–60 | 14 | 11.0 |
| 60–80 | 29 | 8.5 |
| 80–100 | 35 | 15.2 |
| All | 76 | 12.6 |

[1] Number of observations.

For S-1 satellite track 29 ($\theta = 43°10'$), where the incident angle was higher than for track 131 ($\theta = 35°13'$), all of the models gave higher errors of soil moisture estimation. Table 12 presents the mean RMSE errors for both of these tracks separately.

**Table 12.** Errors of soil moisture estimation from developed models for two satellite tracks.

| Orbit/Track | Model 1a | Model 1b | Model 2 |
|---|---|---|---|
| | RMSE (vol. %) | | |
| A/29 | 10.0 | 10.8 | 15.2 |
| A/131 | 9.5 l | 9.9 | 10.7 |

Figures 15 and 16 present a comparison between the soil moisture retrieved by the Model 2 inversion according to Equation (12), and the soil moisture measured at a 5 cm depth by the Decagons GS3 sensors at the grassland and marshland sites. As can be seen in the figures, high compatibility

occurred between the SM values that were modeled and measured; however, it was higher for the marshland site. The lack of response of the Decagon probes to precipitation during the extreme drought in June to September of 2015 can be explained by the hydrophobic effect of the dry peat [39]. The time of reaction of the soil moisture and the retention of water in the soil to precipitation in peat soils is much slower than in mineral soils. After the precipitation that occurred in July and at the beginning of August 2017, the soil moisture has raised in the middle of August at the grassland site and at the end of August at the marshland site.

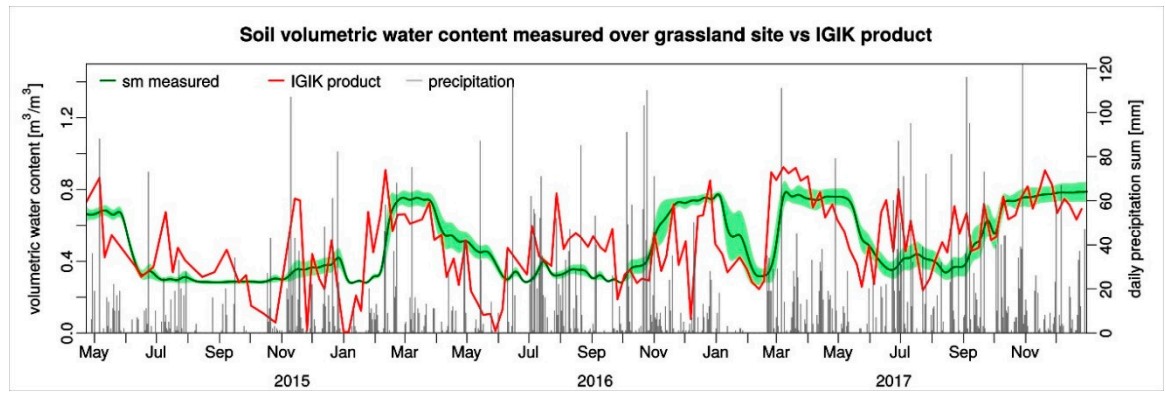

**Figure 15.** Comparison between the soil moisture retrieved by the inversion of Model 2 according to Equation (12) (IGiK (Institute of Geodesy and Cartography) product) and soil moisture measured at a 5 cm depth (sm) by the Decagons GS3 sensors at the grassland site.

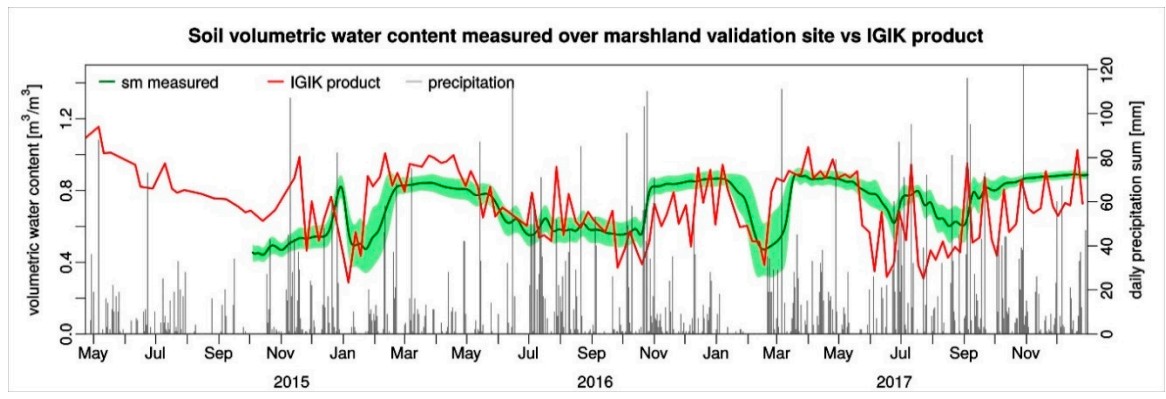

**Figure 16.** Comparison between the soil moisture retrieved by the inversion of Model 2 according to Equation (12) (IGiK (Institute of Geodesy and Cartography) product), and soil moisture measured at a 5 cm depth (sm) by the Decagons GS3 sensors at the marshland site.

The developed model reacts well on the increase of precipitation due to increase of soil moisture and vegetation moisture.

## 4. Discussion

Although previous studies have identified relationships between S-1 $\sigma°$ and the surface soil moisture [16–19,23], this study, for the first time, to our knowledge, in the Biebrza Wetlands, demonstrates the relationships under an extreme range of SM conditions (from dry to wet) i.e., 27–90 vol. %, and different wetland vegetation biomasses (NDVI). The moisture ranges presented, and the diversity of the vegetation biomass, depicts the wetland ecosystems well. The developed models for soil moisture retrieval could be implemented into the system for monitoring areas of wetlands, and in developing decision support and early warning systems.

Two models have been developed based on $\sigma°$ VH and VV, and the NDVI from MODIS. It is evident in Table 6 that for both sites (grassland and marshland) when considered together, the MAPE

errors of σ° as modeled by Model 1a (Equation (7)) and Model 1b (Equation (8)) are comparable; however, for Model 1b, they are slightly higher. Generally, the inversion of the developed σ° models can retrieve the SM with a mean accuracy that is close to 10 vol. %, which is acceptable for the wetland ecosystem authorities and the decision makers. This is especially important for the wetlands areas that are not easily accessible.

The σ° indices as VH−VV and VV/VH, which could replace the vegetation cover as expressed by the NDVI values in soil moisture modeling, have been used to develop Model 2 (Equation (11)). Inversion of Model 2 allows the soil moisture to be retrieved by solely using Sentinel-1 data with a mean accuracy of 13 vol. % (Table 9). Although the accuracy of the soil moisture retrieval using Model 2 was slightly lower than applying Models 1a and 1b, it was still acceptable. Moreover, Model 2 required only microwave data, which is advantageous, especially in areas that are often cloudy.

## 5. Conclusions

The study has shown that the retrieval of soil moisture based on Sentinel-1 data, which considers wetland ecosystems, can be used effectively and with reasonable accuracy (below 10 vol. %). These developments are valuable for areas where in situ data are not available due to the inaccessibility of the area, and when only satellite data can provide suitable tools for decision makers.

The setup of two dense soil moisture measuring networks located over the wetlands offered unprecedented capabilities for modeling the soil moisture from the Sentinel-1 data. The data collected within the study corresponded to from extremely dry (2015) to extremely wet (2017) conditions, which is favorable for the development and validation of soil moisture retrieval models over the wetlands. Also, the selected grassland and marshland sites feature different soil moisture regimes.

Vegetation has to be considered in the relationship between the backscatter and the soil moisture. The vegetation contribution could be expressed by NDVI, or by VV/VH and VH−VV indices that are calculated from the S-1 data.

It has been noted that there is a different contribution of vegetation that is represented by the NDVI when there are dry conditions (SM < 30 vol. %) and moist conditions (SM > 60 vol. %). It was noticed that the values from 50–60 vol. % of soil moisture are within the threshold for the SM influence on σ° VH and VV.

There are discrepancies between Sentinel-1A and Sentinel-1B data. Ascending orbits are better for soil moisture retrieval because the descending overpasses occur during the night when there is dew. The most significant correlation coefficients between the S-1 backscatter and the soil moisture were found for the ascending tracks and for 5 cm depths. A validation was performed for the period of September 2017 until May 2018. The average error was close to 12.6%. It has to be emphasized that the range of the soil moisture in the wetlands was high, at 27–90 vol. %. Such a moisture extent does not occur in agriculture sites. This could also affect the range of the error.

Developed models could be applied for cloudy conditions for sites other than the European Wetlands.

Further work is needed, especially when HH polarization of S-1 is available, to predict the moisture status in wetland ecosystems. The time of reaction of soil moisture and retention of water on precipitation in peat soil was much slower than the reaction to precipitation of other soils. That is why it will be good to examine the time of reaction of SM to precipitation in peat soil.

**Author Contributions:** K.D.-Z. conceived and designed the experiments and wrote the paper; J.M. performed the experiments established the soil moisture network over the B.W. and processed microwave satellite data; A.M. performed statistical analyses; M.B. analyzed the data and contributed in writing the paper; R.G. performed the in situ experiments; W.K. processed and analyzed the optical satellite data; M.B. established the in situ measurements; P.G. performed the database analysis.

**Funding:** The study was funded by ESA EXPRO, contract no. 4000112578/14/NL/MP "Biebrza Sentinel-1 Supersite", and the National project Narodowe Centrum Nauki 2016/23/B/ST10/03155.

**Conflicts of Interest:** The authors declare no conflict of interest.

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
