# Peer review of "Soil Moisture in the Biebrza Wetlands Retrieved from Sentinel-1 Imagery"

_remotesensing, doi:10.3390/rs10121979_

Round 1
Reviewer 1 Report
The proposed paper develop an algorithm for soil moisture estimation over a Wetland area. Different interesting results are presented. However, some aspects linked to discussion of results and methodologies need to be more detailed and clarified.
1) Authors discuss soil moisture estimation, we observe low soil temperature in different periods, what about soil freezing, it has generally important effect on radar signal (see studies with Sentinel1).
2) What is the physical explanation of having a positive slope for figure 4 and a negative slope for figure 5 with the same type of cover !?
3) Could authors add sensitivities of signals to soil moisture in Table 4.
4) Could authors precise the scale of proposed tested radar indices (in dB or linear) for difference of signals VH-VV and also for the ratio.
5) Authors present equations 2, 3, 4 for Cloud Model, and they write that the sigma soil is a linear function of soil moisture, this is the case for dB scale, or in equation (2), sigma soil is in linear scale.
6) What is the relationship between proposed model in equation (6) and WCM, it is not clear to retrieve the proposed model in equation (6) from WCM.
7) Authors illustrate results in figure 7, figure 8, Table 6, what about the validation step for the proposed models , important to clarify calibration and validation steps?
8) Figure 9, how authors explain the signal increase for very high NDVI levels, there is no saturation effect on radar signal, particularly in C-band ?!
9) Table 7, authors retrieve highest errors for low moisture levels, or generally, we retrieve the inverse because of radar signal saturation at high moisture values.
10) Why authors consider MODIS and not Sentinel-2 more adapted to high spatial analysis ?
11) Lines 76-78, authors write : « Atema and Ulaby [8] and Dabrowska et al. [9] proposed a water cloud model…. ». The first reference proposed the model, the second reference applied the model.
12) Lines 120-128, many details should be presented in conclusions and not in introduction.
13) For notation of radar signals in figures and text, authors don’t need to add S1 with Sigma.
14) In the last two years, different new studies have been proposed with C-band data to estimate soil moisture (exemple : Gao et al., 2017, El-Hajj et al., 2017, Tomer et al., 2015 etc), they could be considered in introduction.
Author Response
Dear Reviewer,
Thank you very much for your valuable remarks which have been included in the revised version. Detailed description of corrections made in accordance with your recommendations and suggestion is presented below. Our answers refer to the numbers of figures, tables, and equations included in the corrected manuscript. The first run of the manuscript has been corrected in English by the MDPI English Editing Service (English editing ID: English - 5123).
1) Authors discuss soil moisture estimation, we observe low soil temperature in different periods, what about soil freezing, it has generally important effect on radar signal (see studies with Sentinel1).
The studies covered the period of the growing season (March–October). The effect of freezing the soil did not occur at this time. In the model with only SAR data (Model 2) the measurements with temp<278 K were excluded. It is marked on Fig. 14-15.
2) What is the physical explanation of having a positive slope for figure 4 and a negative slope for figure 5 with the same type of cover !?
Figure 5 presents relationship between the NDVI and s° VH for the high soil moisture conditions (SM >60 vol. %). The negative slope of this relationship is in line with theory - the C-band backscatter is attenuated by increasing vegetation density (the higher NDVI the lower s°). Vegetation plays role in attenuation (two-way attenuation by vegetation) – equations 1-3. In this case sigma is mainly related to t2 s°soil. Figure 6 presents relationship between the NDVI and s° VH for the low soil moisture (SM <30 vol. %). The positive slope of this relationship is due to the dominant influence of total amount of vegetation (the higher NDVI, the higher amount of vegetation). Sigma value is dominated by the backscatter coming from vegetation (s°veg – the first part of the equation 1, and Eq. 2).
3) Could authors add sensitivities of signals to soil moisture in Table 4.
It has been done. Table 4 has been corrected and the sensitivity (as explained in [38]) has been added.
4) Could authors precise the scale of proposed tested radar indices (in dB or linear) for difference of signals VH-VV and also for the ratio.
The scale of tested radar indices is logarithmic, in dB. It was added in the abstract - L25 "... in the logarithmic domain."
5) Authors present equations 2, 3, 4 for Cloud Model, and they write that the sigma soil is a linear function of soil moisture, this is the case for dB scale, or in equation (2), sigma soil is in linear scale.
The description of applying the WCM model was completed and expanded in the 2.4.1 Method section.
6) What is the relationship between proposed model in equation (6) and WCM, it is not clear to retrieve the proposed model in equation (6) from WCM.
The WCM model contains four parameters A-D (described in [37]), which are nonlinearly entangled. Thus, the nonlinear iterative Newton-Raphson method is suitable to solve it. If the B parameter, which is connected to the vegetation cover, is fixed, and the sample contains measurements on quasi bare soil, the equation becomes linear (exactly linearized). The sample of our observation is numerous and precise. It is explained in the text L277: "The following two components of data were designed to describe the effect of the vegetation and the underlying soil on s° VH value: t2 * SM and (1-t2) * cos(q) * NDVI. The first component represents the interaction of the incident radiation between the vegetation and the underlying soil. t2 reduces the impact of the soil on backscatter when the vegetation cover is dense. t2 takes the value from 0–1 and is inversely proportional to the vegetation index and to the incidence angle. The second component describes the remaining part of the backscatter that depends on the vegetation canopy covering the soil. The parameters of the model with s° VH as a dependent variable, and t2 * SM and (1-t2) * cos(q) * NDVI as independent variables, were estimated by applying the Least Squares Regression Method. Data were limited to the vegetation season, i.e. from 60–300 days of each year."
7) Authors illustrate results in figure 7, figure 8, Table 6, what about the validation step for the proposed models , important to clarify calibration and validation steps?
In Table 6 all observations were used, showing the distribution of the error over the location and the tracks. It justifies the joining of the data from two sites and two tracks together. Our final model (Model 2), with radar descriptors of vegetation cover, is supplemented with validation.
8) Figure 9, how authors explain the signal increase for very high NDVI levels, there is no saturation effect on radar signal, particularly in C-band ?!
Figures 10-11 present the simulation of backscatter under different soil moisture and NDVI values. High value of NDVI presents high biomass, and then, high vegetation water content (for wetlands non-forest communities). With the high soil moisture values the curves show the saturation effect. However, better biomass indicator could be LAI. The NDVI does not present well the increase of LAI from some stage of increased biomass.
9) Table 7, authors retrieve highest errors for low moisture levels, or generally, we retrieve the inverse because of radar signal saturation at high moisture values.
The error for different range of soil moisture does not vary much. The signal coming from the soil, when the soil is covered by vegetation, should not saturate and it is not linear relationship.
10) Why authors consider MODIS and not Sentinel-2 more adapted to high spatial analysis ?
Due to the frequent cloud cover over the Biebrza Wetlands we adopted MODIS 8-day compositions in order to analyze soil-vegetation conditions during the whole growing season. Sentinel-2 cloud free images were available in the years 2015-2017 only 5 times during the growing season.
11) Lines 76-78, authors write : « Atema and Ulaby [8] and Dabrowska et al. [9] proposed a water cloud model…. ». The first reference proposed the model, the second reference applied the model.
The sentence has been corrected to: "Atema and Ulaby [8] proposed a water cloud model (WCM) that characterized vegetation as the cloud that represented the total backscatter from the canopy as the sum of the contribution of the vegetation s°veg, and of the underlying soil s°soil. The WCM model was adopted by Dabrowska-Zielinska et al. [9] for agricultural fields."
12) Lines 120-128, many details should be presented in conclusions and not in introduction.
The following sentences have been deleted from Introduction: "It was found that the indices, such as the difference of s° (VH−VV) and the ratio of s° VV/VH, are in monotonic relationships with NDVI. This will give quick information on the soil moisture, using only the Sentinel-1 data. In the developed model, the VV/VH ratio was used as the attenuation factor describing the level of ground and vegetation canopy interaction. The authors used the statistical approach to retrieve soil moisture at grassland and marshland sites taking the time series measurements of in-situ and satellite data. The most significant correlation between backscatter and soil moisture was found for the ascending tracks of Sentinel-1, and for a 5 cm depth of soil moisture."
13) For notation of radar signals in figures and text, authors don’t need to add S1 with Sigma.
Has been deleted.
14) In the last two years, different new studies have been proposed with C-band data to estimate soil moisture (exemple : Gao et al., 2017, El-Hajj et al., 2017, Tomer et al., 2015 etc), they could be considered in introduction.
Line 99 - added the following sentences: "Gao et al. [22] presented two methods for the retrieval of soil moisture over irrigated crop fields based on Sentinel-1 data recorded in the VV polarization combined with Sentinel-2 optical data. The first method used minimum and maximum values of backscattering coefficient calculated from Sentinel-1 data, whereas the second one was based on the analysis of backscattering differences on two consecutive acquisition days. With both methods, the Sentinel-1 data was combined with NDVI index computed from Sentinel-2 data. They obtained estimated RMS soil moisture errors of approximately 0.087 m3m-3 and 0.059 m3m-3 for the first and second methods, respectively. El Hajj et al. [23] used a neural network technique to develop an operational method for soil moisture estimates in agricultural areas based on the synergic use of Sentinel-1 and Sentinel-2 data. They found that VV polarization alone as well as both VV and VH provides better accuracy on the soil moisture calculation than VH alone. The method developed by them could be applied for agricultural plots with an NDVI lower than 0.75 and allows for the soil moisture estimates with an accuracy of approximately 5 vol. %. Baghdadi at al. [24] applied the Water Cloud Model for estimating surface soil moisture of crop fields and grasslands from Sentinel-1/2 data. They simulated the soil contribution (moisture content and surface roughness) applying Integral Equation Model and used NDVI values as the vegetation descriptor. They obtained that the soil contribution to the total radar signal is large in VV polarization when soil moisture is between 5 and 35 vol. %, and NDVI between 0 and 0.8. Tomer et al. [25] developed an algorithm to retrieve surface soil moisture based on the Cumulative Density Function Transformation of multi-temporal RADARSAT-2 backscattering coefficient. The algorithm, which was tested in a semi-arid tropical region in South India and validated with the in-situ data showed RMSE of soil moisture estimates ranging from 0.02 to 0.06 m3m-3 depending on soil information used and development of vegetation."
Added to the References:
22. Gao, Q.; Zribi, M.; Escorihuela, M.J.; Baghdadi, N. Synergetic Use of Sentinel-1 and Sentinel-2 Data for Soil Moisture Mapping at 100 m Resolution. Sensors 2017, 17, 1966, doi:10.3390/s17091966.
23. El Hajj, M.; Baghdadi, N.; Zribi, M.; Bazzi, H. Synergic Use of Sentinel-1 and Sentinel-2 Images for Operational Soil Moisture Mapping at High Spatial Resolution over Agricultural Areas. Remote Sens. 2017, 9, 1292, doi:10.3390/rs9121292.
24. Baghdadi, N.; El Hajj, M.; Zribi, M.; Bousbih, S. Calibration of the Water Cloud Model at C-Band for Winter Crop Fields and Grasslands. Remote Sens. 2017, 9, 969, doi:10.3390/rs9090969.
25. Tomer, S.K.; Al Bitar, A.; Sekhar, M.; Zribi, M.; Bandyopadhyay, S.; Sreelash, K.; Sharma, A.K.; Corgne, S.; Kerr, Y. Retrieval and Multi-scale Validation of Soil Moisture from Multi-temporal SAR Data in a Semi-Arid Tropical Region. Remote Sens. 2015, 7, 8128-8153; doi:10.3390/rs70608128.

Reviewer 2 Report
General Comments
1.The Experimental design to combine Sentinel-1 with NDVI information sounds interesting, but Technical limitation would be that NDVI often provides misleading information of vegetation type and height, failing in reasonably characterizing vegetation.
Decisively, authors confused SAR backscattering with soil mosture. They tried to directly correspond SAR backscattering to in-situ soil moisture measurement. As expected, results are so poor (Tabl,2.3. Figure 4.5.). SAR backscattering is a combination of several factors, only some of which come from surface soil moisture, in this study 50% at best. (Table 2, 3).
.it's poorly written, and poorly organized: Abstract, introduction. This manuscript requires 2nd round review. Main idea should be the first leading sentence, while other supporting sentences will be followed. Authors put the main sentence in the middle of paragraphs and added supporting sentences later. Writing is not readible. grammatically wrong..line by line
3. SM retrieval goal is 5%, but this study shows 10%. This doesn't meet the retrieval goal of Sentinel-1.
Title
how about this: Retrieval of Sentinel-1 surface soil moisture in wetlands by the characterization with NDVI vegetation water contents. You will add a full name of abbreviations.
Abstract
It's poorly written. please rewrite the abstract containing the main issue that you have resolved or the challenge that you wish to tackle, your strategy to approach to that problem, a summary of your method, and results. It's not a brief version of your introduction. In particular, NDVI is very inconsistently placed there. please construct the sentence appropriate to the context.
DO NOT COPY it from introduction. it's same with introduction at L40-45.
Specific comments
L30 What was NDVI condition for that correlation?
L31 Newly developed
L34 specification for Sentinel-1 should be provided in methods. e.g. revisit time of 6 days,
L35 please rephrase the conclusion. the sentence is redundant.
L43. disagree. there are other ways to retrieve soil mositure from satellite: scatterometer, passive microwave.... Rather, please add the merit of SAR.
L45. 6 days of revisit time is high temporal resolution?
L48 please explan why you have chosen VH ,VV
L56 please rephrase changes in soil moisture towards moisture delpletion
L71 please rephrase soil moisture retrieval applying SAR data
L76 by or using
L79 I don't understand this
L105-116 You said wetland is very different from agricultural site at L82, why do you confuse readers by talking about agricultural site here? Are they all vegetations in wetlands?
Vegetations are marshland and grass?
L201 Average-> moderate
L223 pass -> overpass
please specify how much close to each other, 25 min? 15 min?
L226 Sentinel-1 data acquisition time frequency is one day or 6 days?
2.4. Methods
When You said that you use a newly developed inverse approach in abstract, you should provide a thoroughly descried methodology. Description of your method is too short or not sufficiently developed. You skipped all the complex issues arising from your experimental conditions.
It is unclear how you will relate SAR backsacttering with NDVI.
To support your methodology, I think you should emphasize that roughness or NDVI is constant while you correspond SAR backscatering to in-situ soil moisturements. However, it doesn't make sense when you've tested it for three years.
3. Results
there is no consistency in data presentation. Table 4 is for 2015-2016. Figure 6 is 2016. Table 5 is for 2016-2017. Readers may consider this as 'cherry-picking'.
As methodology is wrong - neglecting other factors such as seasonal change in roughness -,
results of Table 2,3 are poor. In Table 4, they seperated data by month. When it is not growing season e.g. March -May, the correlation between SAR and SM is high. When it is growing season e.g.July-September the correlation is very low. This data is somehow reasonable by suppresing other factors having an influence on SAR backscattering so that's why it is relatively fine.
L364-382 should be moved to method section. Please specify in introduction, aim what you want to improve from the traditional water cloud model.
I don't understand how you induce Equation (6)-(12), as SM is not annotated. If it is local SM measurement in the field, there is no point to use this approach with sentinel-1. You should not rely on local measurements. Did you retrieve SM from Sentinel-1? it is not mentioned in method section.
L434 please make a citation for supporting your assumption.
Coefficients in Equation (10) are site-specific.
Although Figure 12, and 13 seem fine, the applicabilitiy of regression approach is restricted by land surface heterogeneity. It means that if readers apply this to other vegetated sites, there is highly uncertain whether it will show similar results with this study.
From equation (6)-(12) please clarify which data e.g. table 2. is used for regression.
Sentinel-1 is satellite. It produces global products. Please describe whether you can apply this approach to the globe where soil moisture local measurements are not available, and you have to find site-specific coefficient in every pixel.
Author Response
Dear Reviewer,
Thank you very much for your valuable remarks which have been included in the revised version. Detailed description of corrections made in accordance with your recommendations and suggestion is presented below. Our answers refer to the numbers of figures, tables, and equations included in the corrected manuscript.
General Comments
1.The Experimental design to combine Sentinel-1 with NDVI information sounds interesting, but Technical limitation would be that NDVI often provides misleading information of vegetation type and height, failing in reasonably characterizing vegetation.
Decisively, authors confused SAR backscattering with soil moisture. They tried to directly correspond SAR backscattering to in-situ soil moisture measurement. As expected, results are so poor (Tabl,2.3. Figure 4.5.). SAR backscattering is a combination of several factors, only some of which come from surface soil moisture, in this study 50% at best. (Table 2, 3).
Table 2 and 3 show that by not considering vegetation cover, the results of direct correlations between backscatter and soil moisture give high errors. That’s why the vegetation cover has to be considered, and was considered.
.it's poorly written, and poorly organized: Abstract, introduction. This manuscript requires 2nd round review. Main idea should be the first leading sentence, while other supporting sentences will be followed. Authors put the main sentence in the middle of paragraphs and added supporting sentences later. Writing is not readible. grammatically wrong..line by line
The first run of the manuscript has been corrected in English by the MDPI English Editing Service (English editing ID: English - 5123).
The article according to the Reviewer recommendation has been reorganized.
3. SM retrieval goal is 5%, but this study shows 10%. This doesn't meet the retrieval goal of Sentinel-1.
The wetland ecosystem is characterized by high soil moisture. The mean SM value at the Biebrza Wetland observed during the study was 63 vol %, however in many cases, the higher moisture appeared. For high soil moisture, the error of 10-13 vol % is acceptable. The decrease or increase of 20 vol % in soil moisture is still important information for the decision makers.
Title
how about this: Retrieval of Sentinel-1 surface soil moisture in wetlands by the characterization with NDVI vegetation water contents. You will add a full name of abbreviations.
The NDVI was applied to Models 1A and 1B and to check if it can be replaced with the indices VH-VV and VV/VH. In the developed model 2 the NDVI has been replaced with the indices VH-VV and VV/VH. This approach allows using Sentinel-1 data alone, and the title may remain unchanged.
Abstract
It's poorly written. please rewrite the abstract containing the main issue that you have resolved or the challenge that you wish to tackle, your strategy to approach to that problem, a summary of your method, and results. It's not a brief version of your introduction. In particular, NDVI is very inconsistently placed there. please construct the sentence appropriate to the context.
DO NOT COPY it from introduction. it's same with introduction at L40-45.
The abstract has been change to:
"The objective of the study was to estimate soil moisture (SM) from Sentinel-1 (S-1) satellite images acquired over wetlands. The study was carried out during the years 2015–2017 in the Biebrza Wetlands, situated in northeastern Poland. At the Biebrza Wetlands, two Sentinel-1 validation sites were established, covering grassland and marshland biomes, where a network of 18 stations for soil moisture measurement was deployed. The sites were funded by the European Space Agency (ESA), and the collected measurements are available through the International Soil Moisture Network (ISMN). The SAR data of the Sentinel-1 satellite with VH (vertical transmit and horizontal receive) and VV (vertical transmit and vertical receive) polarization were applied to SM retrieval for a broad range of vegetation and soil moisture conditions. The methodology is based on research into the effect of vegetation on backscatter (s°) changes under different soil moisture and Normalized Difference Vegetation Index (NDVI) values. The NDVI was derived from the optical imagery of a MODIS (Moderate Resolution Imaging Spectroradiometer) sensor onboard the Terra satellite. It was found that the state of the vegetation expressed by NDVI may be described by the difference between s° VH and VV, or the ratio of s° VV/VH, as calculated from the Sentinel-1 images in the logarithmic domain. The most significant correlation coefficient for soil moisture was found for data that was acquired from the ascending tracks of the Sentinel-1 satellite, characterized by the lowest incidence angle, and SM at a depth of 5 cm. The study demonstrated that the use of the inversion approach, which was applied to the new developed models and includes the derived indices based on S-1, allowed the estimation of SM for peatlands with reasonable accuracy (RMSE ~ 10 vol. %). The developed soil moisture retrieval algorithms based on S-1 data are suited for wetland ecosystems, where soil moisture values are several times higher than in agricultural areas."
Specific comments
L30 What was NDVI condition for that correlation?
NDVI is not considered for the Model 2.
L31 Newly developed
Has been changed.
L34 specification for Sentinel-1 should be provided in methods. e.g. revisit time of 6 days,
The temporal frequency of the two S-1 satellites is included in paragraph 2.3.
L35 please rephrase the conclusion. the sentence is redundant.
The sentence "The conclusion drawn from the study emphasizes a demand for the derivation of specific soil moisture retrieval algorithms that are suited for wetland ecosystems, where soil moisture is several times higher than in agricultural areas." has been change to: "The developed SAR soil moisture retrieval algorithms based on S-1 data are suited for wetland ecosystems, where soil moisture values are several times higher than in agricultural areas".
L43. disagree. there are other ways to retrieve soil moisture from satellite: scatterometer, passive microwave.... Rather, please add the merit of SAR.
The sentence "The SAR satellite imagery is the only mean to fulfill this objective regardless of cloud cover and, especially in the areas, in which deployment of in-situ SM measurements is not possible or economically unprofitable." has been change to: "The SAR satellite imagery is an important source to fulfill this objective regardless of cloud cover and, especially in the areas, in which deployment of in-situ SM measurements is not possible or economically unprofitable"
L45. 6 days of revisit time is high temporal resolution?
Yes, for soil moisture monitoring in wetlands.
L48 please explan why you have chosen VH ,VV
There is no other polarization available from S-1 for this area.
L56 please rephrase changes in soil moisture towards moisture delpletion
The sentence "Changes in soil moisture towards moisture depletion cause changes in the soil, and the release of substantial amounts of carbon into the atmosphere [2, 3]" has been change to: "Changes in soil moisture towards depletion cause peat mineralization, and the release of substantial amounts of carbon into the atmosphere [2, 3]"
L71 please rephrase soil moisture retrieval applying SAR data
The sentence "The advances in soil moisture retrieval applying SAR data described Kornleson and Coulibaly [5]" has been change to: "Kornleson and Coulibaly [5] conducted a comprehensive literature review to provide soil moisture retrieval methodology from SAR data."
L76 by or using
The sentence "The strong interactions of the backscatter signal with the soil and vegetation may not be expressible by using simple linear functions." has been change to: "The strong interactions of the backscatter signal with the soil and vegetation may not be expressible by a simple linear functions."
L79 I don't understand this
The sentence "The separation of vegetation that is influenced by the soil moisture by the received microwave signals is not straightforward." has been change to: "The separation of the soil and vegetation components is not straightforward due to complex interactions between them that simultaneously affect SAR backscatter."
L105-116 You said wetland is very different from agricultural site at L82, why do you confuse readers by talking about agricultural site here? Are they all vegetations in wetlands?
Vegetations are marshland and grass?
The sentences in L105-116: "Santi et al. [22] carried out an investigation in an agricultural area located in North-west Italy, and found that the soil moisture values retrieved from the C-band ENVISAT/ASAR simulated by a hydrological model, and the measured values in situ, were in good agreement." and "Vreugdenhil et al. [24] examined the sensitivity of Sentinel-1 to vegetation dynamics and examined VV and VH backscatter and their ratio VH/VV to monitor crop conditions with special reference to vegetation water content (VWC) of agriculture crop. Greifeneder et al. [25] also analyzed the added value of the ratio of VH/VV for soil moisture estimates and demonstrated that the ratio of VH/VV allows a good compensation of vegetation dynamics for the retrieval of soil moisture". have been removed.
L201 Average-> moderate
"Average" has been changed to "moderate"
L223 pass -> overpass
please specify how much close to each other, 25 min? 15 min?
The sentence "The pass times of Sentinel-1 and Terra MODIS (8-day compositions)... " has been changed to "The acquisition dates of Sentinel-1 and Terra MODIS (8-day compositions)... "
L226 Sentinel-1 data acquisition time frequency is one day or 6 days?
The sentence "MODIS 8-day data was smoothed to a one-day time series, and the data was synchronized with the Sentinel-1 date of acquisition." has been changed to: "MODIS NDVI 8-day compositions were paired with Sentinel-1 acquisition dates."
2.4. Methods
When You said that you use a newly developed inverse approach in abstract, you should provide a thoroughly descried methodology. Description of your method is too short or not sufficiently developed. You skipped all the complex issues arising from your experimental conditions.
It is unclear how you will relate SAR backsacttering with NDVI.
It has been written in section 2.3 L247.
To support your methodology, I think you should emphasize that roughness or NDVI is constant while you correspond SAR backscatering to in-situ soil moisturements. However, it doesn't make sense when you've tested it for three years.
The section Method 2.4 was developed.
3. Results
there is no consistency in data presentation. Table 4 is for 2015-2016. Figure 6 is 2016. Table 5 is for 2016-2017. Readers may consider this as 'cherry-picking'.
Table 4 has been corrected and results were described.
As methodology is wrong - neglecting other factors such as seasonal change in roughness -,
results of Table 2,3 are poor. In Table 4, they seperated data by month. When it is not growing season e.g. March -May, the correlation between SAR and SM is high. When it is growing season e.g.July-September the correlation is very low. This data is somehow reasonable by suppresing other factors having an influence on SAR backscattering so that's why it is relatively fine.
In April-May there is the growing season at wetlands with the increase of NDVI (Fig. 7). Following Fig. 7, the growing season at wetlands continue (vegetation development). Table 4 presents the correlation between soil moisture and backscatter for each considered month. The changes in NDVI also seasonal were considered in the developed models 7-8. In March the vegetation conditions were close to bare soil.
L364-382 should be moved to method section.
The description of WCM with the Least Square Regression Method has been moved to 2.4 Methods as subsection 2.4.1.
Please specify in introduction, aim what you want to improve from the traditional water cloud model.
Has been done.
I don't understand how you induce Equation (6)-(12), as SM is not annotated. If it is local SM measurement in the field, there is no point to use this approach with sentinel-1. You should not rely on local measurements. Did you retrieve SM from Sentinel-1? it is not mentioned in method section.
Description of method has been developed and supplied with additional explanations.
L434 please make a citation for supporting your assumption.
This phrase has been changed.
Coefficients in Equation (10) are site-specific.
Although Figure 12, and 13 seem fine, the applicabilitiy of regression approach is restricted by land surface heterogeneity. It means that if readers apply this to other vegetated sites, there is highly uncertain whether it will show similar results with this study.
From equation (6)-(12) please clarify which data e.g. table 2. is used for regression.
The goal of the research was to estimate the soil moisture based on S-1 data. Therefore in Eq. 12 the only data used were: sigma VH-VV and sigma VH/VV. As the Fig 14 and 15 present the soil moisture may be calculated for the days of S-1 acquisitions i.e. every 6 days (applying s° VH and VV data). It is written in L325 that ascending orbits are used for further analysis. Besides, it has been added at the end of L423 and L443 "for ascending orbits".
Sentinel-1 is satellite. It produces global products. Please describe whether you can apply this approach to the globe where soil moisture local measurements are not available, and you have to find site-specific coefficient in every pixel.
The methodology may be applied to other sites which are peatlands at the wetlands (non-forest communities) ecosystems.

Reviewer 3 Report
Summary and Review Comments:
The backscattering of SAR is affected by two components of the land surface, i.e. vegetation and soil. With the Sentinel-1 SAR backscattering in VH and VV, the authors tested out the use of the classic vegetation index NDVI and the difference and ratio of the two polarizations, i.e. VV and VH, as the proxy for vegetation in the water cloud model. Model inversions of the two abovementioned cases were performed to retrieve soil moisture at two wetland sites, one grassland and the other marshland, with RMSE of ~10%.
Specific suggestions are listed below:
1) While the manuscript itself is complete and comprehensive, the abstract needs a bit more polishing. Below are some suggestions that may be of help: Use one or two sentences, briefly describe the main idea of the new methodology, especially how NDVI was used in the algorithm. More focus would be placed on the use of NDVI, the difference in VV and VH and the ratio of the two polarizations as the proxy for vegetation conditions, as well as the advantages of this approach.
2) L49-56: the description of the study sites is more suitable for Section 2.1
3) L69-84: Not as an expert of wetland, I cannot help to wander that, in comparison to cropland, what are the additive challenges posed by the wetland ecosystem in terms of the retrieval of soil moisture?
4) L225: Please give more details about the NDVI processing. The original MODIS NDVI should be prone to noise, thus low values can be often seen in the seasonal trajectory. I guess the authors used the quality flag to rule out the low-quality NDVI or NDVI retrieved from the backup algorithm in the beginning? And then how the 8-day time series was ‘smoothed’ to daily interval?
5) Is there any specific reason why the ratio vv/vh was used in the formulation of Eq. (9) while (vh-vv)2 was used in Eq. (10)? Did the authors tested out different combinations and found out that this combination outperforms the others?
6) Please also add some theory basis of why the difference and ratio of the two polarizations can be used as the proxy of vegetation in the manuscript.
7) Please also show the R2 and scatter plots of data corresponding to Table 11 for Model 2 and Tables 7 and 8 for Model 1a. According to Fig. 12, the inversion approach tends to underestimate when volumetric soil moisture is ~40%, e.g. Jun.-Sept. 2016, for the grassland site, and underestimate when soil moisture goes higher.
8) Talking about the overestimation of soil moisture for the grassland site in Fig. 12 (results from Model 2), I begin to wander if there is also such a problem with Model 1 using NDVI as the proxy for vegetation. NDVI during this period is over 0.8 according to Fig. 6, which fails to respond to the increase in leaf area index. This notorious issue of NDVI saturation with high LAI could make the vegetation component underrepresented in the model, and therefore overestimation of soil moisture when soil is relatively dry.
9) Last but not the least, please read through the manuscript to screen out grammatical errors or sentence fragments for correction, e.g. L71, L79, and etc.
Overall, other than some polishing of the manuscript, I recommend accept with minor revision.
Author Response
Dear Reviewer,
Thank you very much for your valuable remarks which have been included in the revised version. Detailed description of corrections made in accordance with your recommendations and suggestion is presented below. Our answers refer to the numbers of figures, tables, and equations included in the corrected manuscript. The first run of the manuscript has been corrected in English by the MDPI English Editing Service (English editing ID: English - 5123).
1) While the manuscript itself is complete and comprehensive, the abstract needs a bit more polishing. Below are some suggestions that may be of help: Use one or two sentences, briefly describe the main idea of the new methodology, especially how NDVI was used in the algorithm. More focus would be placed on the use of NDVI, the difference in VV and VH and the ratio of the two polarizations as the proxy for vegetation conditions, as well as the advantages of this approach.
Changes in L246: "The area of an SM sensors sites is 500 × 500 m. The soil moisture, s°, and NDVI were taken as the average values for this area."
It was proved that NDVI may be replaced by the vegetation descriptors such as indices VH-VV and VV/VH (Fig. 7, Fig. 13, Tab. 5).
2) L49-56: the description of the study sites is more suitable for Section 2.1
The sentences in Introduction "The Biebrza Wetlands holds 25,494 ha of peatlands, much biodiversity in the rich plant habitats, as well as highly diversified fauna, especially for birds [1]. This is still one of the wildest areas in Europe, and one of the areas that has been least destroyed, damaged, or changed by human activity." have been moved to Section 2.1 L156.
3) L69-84: Not as an expert of wetland, I cannot help to wander that, in comparison to cropland, what are the additive challenges posed by the wetland ecosystem in terms of the retrieval of soil moisture?
The main difference is in soil type and level of soil moisture. Agricultural soils are mainly mineral, while wetland soils are organic - peaty. These soil diferences cause soil moisture diferences. SM<30 vol. % in wetlands is extremely low and leads to drying of peat soils and its mineralization. Monitoring of SM at wetlands is very important due to SM variations, where depletion cause peat mineralization, and the release of substantial amounts of carbon into atmosphere [2,3]. In arable lands SM of about 30 vol. % is extremely high and leads to soaking the plants. Also, the vegetation conditions in both ecosystems in the same soil moisture conditions (30 vol. %) differ significantly. Thus, the models developed for crops or bare soils could not be adopted for wetlands.
4) L225: Please give more details about the NDVI processing. The original MODIS NDVI should be prone to noise, thus low values can be often seen in the seasonal trajectory. I guess the authors used the quality flag to rule out the low-quality NDVI or NDVI retrieved from the backup algorithm in the beginning? And then how the 8-day time series was ‘smoothed’ to daily interval?
We used MODIS NDVI Product MOD09Q1 latest version V006 with improvements of aerosols retrieval for atmospheric correction and changes in snow/cloud/cloud shadow detection algorithms. Product MOD09Q1 is NDVI eight-day composition is prone to noise and smoothed on the basis of eight daily observations. We used quality flags, snow/cloud/cloud shadow in order to rule-out the low quality NDVI (especially in the beginning of the growing season). MODIS NDVI 8-day compositions were paired with Sentinel-1 acquisition dates.
The sentences in L223-228 (old version) have been change to: " The acquisition dates of Sentinel-1 and Terra MODIS (8-day compositions) were close to each other; therefore, it was assumed that NDVI values could be used to represent the vegetation effect for the modeling of the backscattering coefficients of the S-1. The area of an SM sensors sites is 500 × 500 m. The soil moisture, s° and NDVI were taken as the average values for this area." - new L44-247.
5) Is there any specific reason why the ratio vv/vh was used in the formulation of Eq. (9) while (vh-vv)2 was used in Eq. (10)? Did the authors tested out different combinations and found out that this combination outperforms the others?
There are many publications where the authors cite the results showing that the CR ratio (VV/VH) is applied to monitor vegetation dynamics, crop condition, crop classification. In our experiment the values of VH and VV were always negative and |VV| < |VH|. The index VV/VH indicates the magnitude of scattering by vegetation to vegetation and soil moisture. In Model 2 the index VV/VH is used in the attenuation part describing the level of ground and vegetation canopy interaction (Eq. 10). The model uses the SAR-derived description of vegetation in WCM, s° (VH−VV). Table 5 and Fig. 7 proved the relationship between NDVI and vegetation descriptors as VH-VV and VV/VH. Figure 13 shows the periods under low vegetation conditions. For our sites, S-1 has only VV and VH polarizations. The proposed combinations were considered as the best.
6) Please also add some theory basis of why the difference and ratio of the two polarizations can be used as the proxy of vegetation in the manuscript.
The possibility to use the difference and ratio of the two polarizations (indices: VH-VV and VV/VH) as the proxy of vegetation conditions has been demonstrated by relationships that were found between NDVI and both indices. Application of the indices VH-VV and VV/VH is compatible with the results when the NDVI data were used (Fig. 10-11). For low SM there is the increase of s° VH. For high values of SM, there is the attenuation of the beam by vegetation L523-525. Besides, several examples of the use of these indices can be found in the literature, which could be added to the main text. We propose some of them - see below.
The following sentences have been added in L127:
"The difference and the ratio of the VH and VV backscatter as the proxy of vegetation conditions has been recently studied and published by several researchers. Vreugdenhil et al. [27] examined Sentinel-1 VV and VH backscatter and their ratio VH/VV to monitor crop conditions with special reference to vegetation water content (VWC) of agriculture crop. Greifeneder et al. [28] demonstrated that the ratio of VH/VV calculated from AQUARIUS L-band scatterometer allows a good compensation of vegetation dynamics for the retrieval of soil moisture. Hosseini et al. [29] used RADARSAT-2 to estimate Leaf Area Index (LAI) for corn and soybeans fields. They found high correlation coefficients between ground measured and estimated LAI values,when dual like-cross polarizations were used (either HH–HV or VV–HV). Also, it has been found that RADARSAT-2 (HH-HV) can be used for the retrieval of soil moisture and the total biomass, while RADARSAT-2 (VV-HV) can be used for the retrieval of the biomass of the wheat heads [30]."
Added references:
[27] Vreugdenhil, M.; Wagner, W.; Bauer-Marschallinger, B.; Pfeil, I.; Teubner, I.; Rüdiger, C.; Strauss, P. Sensitivity of Sentinel-1 Backscatter to Vegetation Dynamics: An Austrian Case Study. Remote Sens. 2018, 10, 1396, DOI:10.3390/rs10091396.
[28] Have been previously included.
[29] Hosseini, M.; McNairn, H.; Merzouki, A.; Pacheco, A. Estimation of Leaf Area Index (LAI) in corn and soybeans using multi-polarization C- and L-band radar data. Remote Sens. Environ., 2015, 170, 77–89, http://dx.doi.org/10.1016/j.rse.2015.09.002.
[30] Hosseini, M. and McNairn, H. Using multi-polarization C- and L-band synthetic aperture radar to estimate biomass and soil moisture of wheat fields. Int. J. Appl. Earth Observation and Geoinformation, 2017, 58, 50–64, http://dx.doi.org/10.1016/j.jag.2017.01.006
7) Please also show the R2 and scatter plots of data corresponding to Table 11 for Model 2 and Tables 7 and 8 for Model 1a. According to Fig. 12, the inversion approach tends to underestimate when volumetric soil moisture is ~40%, e.g. Jun.-Sept. 2016, for the grassland site, and underestimate when soil moisture goes higher.
The scatterplots present Fig. 1-3 below:
Table 11 presents the results of the validation procedure for the year 2018.
Fig. 1. Scatterplot for Table 11.
Fig. 2. Scatterplot for Table 9 – Modeled values of SM against measured SM for for the data above 278 K.
Fig. 3. Scatterplot for Table 7-8. Model with the optical data for vegetation from the day 60 to 300
The underestimated values of modeled SM (Fig. 15) occurred in most cases in November, December, and January, when the soil moisture values (measured) are not stable.
8) Talking about the underestimation of soil moisture for the grassland site in Fig. 12 (results from Model 2), I begin to wander if there is also such a problem with Model 1 using NDVI as the proxy for vegetation. NDVI during this period is over 0.8 according to Fig. 6, which fails to respond to the increase in leaf area index. This notorious issue of NDVI saturation with high LAI could make the vegetation component underrepresented in the model, and therefore overestimation of soil moisture when soil is relatively dry.
One of the goals of the study was to replace optical data with the SAR data only.
This approach is important for the future, where there is a lack of optical data.
Figure 14 proves the compatibility of SAR indices with NDVI.
9) Last but not the least, please read through the manuscript to screen out grammatical errors or sentence fragments for correction, e.g. L71, L79, and etc.
Overall, other than some polishing of the manuscript, I recommend accept with minor revision.

Round 2
Reviewer 1 Report
The authors answer all comments. Minor corrections are proposed to improve the paper.
1) The quality of the figures could be improved, it would be necessary to homogeinize the formats, with or without borders, with or without grids etc,
2) In equation (2), the power E seems to me a writing error.
Author Response
Dear Reviewer,
Thank you very much for your comments. All of them have been taken into account.
The authors answer all comments. Minor corrections are proposed to improve the paper.
1) The quality of the figures could be improved, it would be necessary to homogeinize the formats, with or without borders, with or without grids etc,
All Figures without borders and grids have been corrected. All Figures are in required quality and loaded separatelly as zip file.
2) In equation (2), the power E seems to me a writing error.
In L265 E has been added as fitted parameter. In our modified WCM we have used the Least Squares Method where E has not been considered (Eq. 4). Many authors working with WCM have not considered "E" ex. [24].
Submission Date
19 October 2018
Date of this review
22 Nov 2018 09:00:52

Reviewer 3 Report
Thank you for the responses. However, some of the explanations were expected to be reflected in the manuscript, e.g. Q4 in the previous comments regarding the processing of NDVI.
My biggest concern of the manuscript now is that this study relies heavily on the use of SAR indices calculated from the backscattering from the co- and cross-polarizations, i.e. vv and vh. Although after revision, the authors added more reference on successfully relating the two backscattering values to the dynamics of vegetation, there is a still lack of mechanistic explanation on why the ratio/difference can be used as the proxy of vegetation. This can be done briefly in the introduction or more comprehensively in the discussion section.
I see the point that the newly added Fig. 13 is trying to make, i.e. show the temporal evolution of the ratio and the difference. But the authors may consider combining Fig. 13 with Fig. 7 due to the similarity of the contents of the two figures. Also, there could be a mess up with the data: as stated in the manuscript, Fig. 7 and Fig. 13 both showed the grassland site in 2016, but why the vh-vv in two figures are different? Mistakenly showing the other site in one of the figures?
Fig. 15, the overestimation of soil moisture during Jul. and Sept., 2016 is worthy of further discussion in Section 4 (This was typed wrongly in my previous comments Q 7&8 as ‘underestimation’, but actually I meant ‘overestimation’ as stated in the end of Q8.). No only this period, overestimation usually occurs when the measured sm is lower than 40%. Based on Figs. 6 and 14 showing Sigma vh increases with increasing vegetation proxies over drier soils, the overestimation of SM in Fig. 15 could probably be explained in terms of the misrepresentation of vegetation by the proxy.
Author Response
Dear Reviewer,
Thank you very much for your comments. Detailed description of corrections made in accordance with your recommendations is presented below.
Thank you for the responses. However, some of the explanations were expected to be reflected in the manuscript, e.g. Q4 in the previous comments regarding the processing of NDVI.
We downloaded MODIS scenes from MOD09Q1 version 6 (V006) data product. MOD09Q1 V006 (DOI: 10.5067/MODIS/MOD09Q1.006) provide an estimate of the surface spectral reflectance of Terra MODIS bands (cantered at 645 nm and 858 nm) at a 250 m resolution in a 8-day gridded level-3 product in the sinusoidal projection. The not clouded observation for the 8-day period, for the area was taken. For each pixel, a value is selected from all the acquisitions within the 8-day composite on the basis of high observation coverage, low view angle, the absence of clouds or cloud shadow, and aerosol loading. We calculated NDVI from MODIS bands 1 and 2 as follows NDVI=(band2-band1)/(band2+band1). For calculating NDVI we take the spectral reflectance values larger than 0 and lower than 10000 (16 bit unsigned integer). All pixels with values lower than 0 and bigger than 10000 are not taken. Full equation of calculating NDVI is presented as follows:
Equation = 'UINT((((float(b1) - float(b2))/(float(b1) + float(b2))) * 10000 + 10000) * ((b1 GT 0 AND b1 LT 10000 AND b2 GT 0 AND b2 LT 10000)) + ((b1 LE 0 OR b1 GE 10000 OR b2 LE 0 OR b2 GE 10000)))'
We also exploited the band 3 entitled Surface Reflectance 250m State flags, from which we extracted pixels flagged as water, clouds/cloud shadows and snow/ice. Firstly, we applied water mask on NDVI images, next we overlaid snow/ice mask for the image NDVI with recognized water. The last step concerns applying cloud mask containing clouds and cloud shadows pixels on the NDVI image with water and snow/ice masks overlaid. The NDVI images are distributed as digital numbers in 16 bit unsigned integer format.
In L245 the sentence: "The acquisition dates of Sentinel-1 and Terra MODIS (8-day compositions) were close to each other;" has been change to: "MODIS NDVI 8-day compositions were paired with Sentinel-1 daily satellite images, so that the nearest day of S-1 acquisition to the middle date of 8-day composition of MODIS was taken."
In L304 the following text has been added:
"For calculating NDVI all pixels with the spectral reflectance values larger than 0 and lower than 10000 (16 bit unsigned integer) were taken. Then, from Band 3 (Surface Reflectance 250m State flags) of MOD09Q1 product the pixels flagged as: water, clouds/cloud shadows, and snow/ice were extracted and applied to NDVI images."
My biggest concern of the manuscript now is that this study relies heavily on the use of SAR indices calculated from the backscattering from the co- and cross-polarizations, i.e. vv and vh. Although after revision, the authors added more reference on successfully relating the two backscattering values to the dynamics of vegetation, there is a still lack of mechanistic explanation on why the ratio/difference can be used as the proxy of vegetation. This can be done briefly in the introduction or more comprehensively in the discussion section.
I see the point that the newly added Fig. 13 is trying to make, i.e. show the temporal evolution of the ratio and the difference. But the authors may consider combining Fig. 13 with Fig. 7 due to the similarity of the contents of the two figures. Also, there could be a mess up with the data: as stated in the manuscript, Fig. 7 and Fig. 13 both showed the grassland site in 2016, but why the vh-vv in two figures are different? Mistakenly showing the other site in one of the figures?
In L-146 the new sentence has been added: "Application of these descriptors as dual polarisation give better results to separate the influence of vegetation from the soil moisture impact on backscatter."
In Figure 7 there is a comparison of one growing season course of NDVI and sigma VH-VV on grassland site and satellite track 29 as an example of monothonic relation between them. In Figure 13 the whole two years 2016-2017 of courses the sigma VV/VH and sigma VH-VV are shown. The goal is to compare the two vegetation proxies, catch the minimal values of sigma VV/VH and point out that it takes place in early spring periods when the vegetation is low. Data used in Figure 13 are from grassland site 2016, but from other satellite track i.e. 131 (not 29 like in Figure 7). That is the reason of the difference between the two graphs. We've prepared a new version of Figure 13 with satellite track 29 as in Figure 7.
Figure 13 has been change and new version is included. Figure 13 has been replaced with Figure 14 to which we refer earlier in the L511.
Fig. 15, the overestimation of soil moisture during Jul. and Sept., 2016 is worthy of further discussion in Section 4 (This was typed wrongly in my previous comments Q 7&8 as ‘underestimation’, but actually I meant ‘overestimation’ as stated in the end of Q8.). No only this period, overestimation usually occurs when the measured sm is lower than 40%. Based on Figs. 6 and 14 showing Sigma vh increases with increasing vegetation proxies over drier soils, the overestimation of SM in Fig. 15 could probably be explained in terms of the misrepresentation of vegetation by the proxy.
The overestimation of modelled soil moisture in relation to measured occurred in July-September of 2016 (Fig.15). The measured soil moisture values indicate 40%. Considering high precipitation at this time, the measured values seem to be too low. The model reacts very well what could be caused also by the wetness of the vegetation. At the marchland, there is no such discrepancy.
In lines L479 and L521 behind the "track 29" the following part of text has been added in brackets "(q = 43° 10')".
In lines L481 and L523 behind the "track 131" the following part of text has been added in brackets "(q = 35° 13')".
In line L571 the following sentence has been adde: " The developed model reacts well on the increase of precipitation due to increase of soil moisture and vegetation moisture "
Submission Date
19 October 2018
Date of this review
23 Nov 2018 21:50:35
